# Value Gradient Guidance for Flow Matching Alignment

**Zhen Liu**[1†]    **Tim Z. Xiao**[2*]    **Carles Domingo-Enrich**[3*]    **Weiyang Liu**[4]    **Dinghuai Zhang**[3,5†]

[1]The Chinese University of Hong Kong (Shenzhen)    [2]University of Tübingen
[3]Microsoft Research    [4]The Chinese University of Hong Kong    [5]Mila – Quebec AI Institute
[*]Equal contribution    [†]Corresponding author    `vgg-flow.github.io`

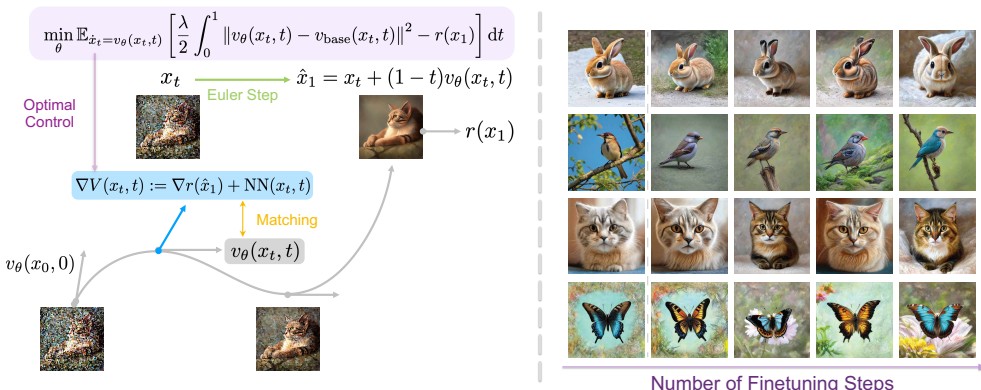

Figure 1: Left: Illustration of our proposed **VGG-Flow** algorithm. The velocity field is trained to **match** the **value gradient** field obtained from the **optimal control** problem. The value gradient field is parametrized as the reward gradient field of the **one-step Euler prediction** plus a learnable residual field. Right: Evolution of samples (with fixed seeds and prompts) during the course of finetuning on the reward model of Aesthetic Score.

## Abstract

While methods exist for aligning flow matching models – a popular and effective class of generative models – with human preferences, existing approaches fail to achieve both adaptation efficiency and probabilistically sound prior preservation. In this work, we leverage the theory of optimal control and propose **VGG-Flow**, a gradient-matching–based method for finetuning pretrained flow matching models. The key idea behind this algorithm is that the optimal difference between the finetuned velocity field and the pretrained one should be matched with the gradient field of a value function. This method not only incorporates first-order information from the reward model but also benefits from heuristic initialization of the value function to enable fast adaptation. Empirically, we show on a popular text-to-image flow matching model, Stable Diffusion 3, that our method can finetune flow matching models under limited computational budgets while achieving effective and prior-preserving alignment.

## 1   Introduction

Flow matching models [1, 36, 38] are one of the most effective methods in modeling high-dimensional real-world continuous distributions and widely used for the generation of images [17], videos [65], 3D objects [39, 40, 69, 78], etc. These models, compared to diffusion models that rely on simulation with stochastic differential equations (SDEs), are trained to sample with deterministic ordinary differential equations (ODEs) of which sampling paths are often straighter and easier to model.

Similar to the motivations for performing alignment for diffusion models [6, 19], it is natural to finetune flow matching models with reward models so that the generated samples are more aligned

39th Conference on Neural Information Processing Systems (NeurIPS 2025).

with human preferences. While existing methods have already achieved fast, effective, diversity-preserving and prior-preserving alignment for diffusion models through gradient-matching-based approaches, the ODE sampling paths of flow matching models pose challenges in applying these methods. The key challenge is that, with flow matching models, one typically has access to neither a reference path (unless one has access to the large-scale pretraining dataset) nor the probability flow. Since it is non-trivial to obtain the probability flow and to incorporate the learned prior from base models for flow matching models, it is harder to align flow matching models in an efficient yet probabilistic way.

To address this issue, we take inspiration from the theory of optimal control and consider a relaxed objective: we optimize the target reward but with the accumulated cost-to-go defined as the $\ell_2$ distance between the velocity fields of the finetuned model and the base model. The optimal solution of this optimization program is described by the Hamilton-Jacobi-Bellman (HJB) equation and can be shown in our formulation equivalent to two conditions: a gradient matching condition that the residual velocity field matches the gradient of the value function, and a value consistency condition that ensures correct estimation of value functions. In light of this result, we propose our finetuning method, dubbed **VGG-Flow** (short for **V**alue **G**radient **G**uidance for **Flow** Matching Alignment), that finetunes the flow matching model via "matching with value gradient guidance"—the difference between the velocity fields of the finetuned model and the base model is expected to be the gradient of the value function—while the value function can be solved with a consistency equation. Such a formulation allows us not only to directly propagate the reward gradient to the matching target through the value consistency equation in an amortized and memory-efficient way but also to use a heuristic initialization of the value gradient for fast convergence. We empirically show that **VGG-Flow** can effectively and robustly finetune large flow matching models like Stable Diffusion 3 [17] within limited computational resources.

To summarize, our contributions are

- With a relaxed objective, we leverage the HJB equation from optimal control theory to propose **VGG-Flow**, an efficient and effective alignment method for flow matching models that matches the residual velocity field with the guidance signal of value function gradient.

- We propose to parametrize the value gradients with a forward-looking technique, which eases the difficulty in learning accurate value gradients in limited time and thus accelerates convergence.

- We empirically demonstrate the effectiveness of **VGG-Flow** on a large-scale text-to-image flow matching model, Stable Diffusion 3, and show that **VGG-Flow** achieves better reward convergence, sample diversity, and prior preservation compared to other alignment baselines.

## 2 Related Work

**General alignment strategies.** Since large generative models are typically trained on uncurated massive datasets, their sample distributions are typically far from human preferences. A common approach to solve this problem is through reinforcement learning from human feedback (RLHF) [45], in which one first trains a reward model from human preference datasets and later finetunes a generative model with reinforcement learning methods such that it samples from this reward model. In the alignment of large language models, it is common to use simple policy gradient methods such as PPO [56] and GRPO [57]. While they are general enough to be applicable to continuous generative models, they can be less efficient because they do not leverage the differentiable nature of both reward models and generative models. Similar to traditional RL [73] methods, the framework of generative flow networks (GFlowNets) [4, 46, 74, 75, 76], which are highly correlated with soft RL methods, can be used to finetune diffusion models [41, 71, 72, 77]. Alternatively, one may simple reward-reweight methods [18, 19, 34] for the same purpose. It is also under exploration to perform test-time scaling on diffusion models via methods like sequential Monte Carlo [27, 58], parallel tempering [28], and search [42] without any model finetuning.

**Differentiable RLHF for continuous foundation generative models.** Diffusion models and flow matching models, commonly used to build foundation models in the continuous domain, exhibit different properties due to their differentiable and sequential sampling process. For diffusion models, since each sampling step is stochastic (probably excluding the last step), one may finetune these models using stochastic optimal control [61] which typically requires extra steps for learning surrogate functions. A recent work inspired by the framework of generative flow networks [41] builds a gradient-informed finetuning strategy that efficiently aligns diffusion models with gradient-matching-like

losses in a probabilistic way. For flow matching models, these approaches are not applicable because they require the transitions to be stochastic. One way that applies to both diffusion models and flow matching models is to treat the sampling process as a computational graph with which we directly optimize differentiable rewards [12, 70]. However, such a strategy by design fails to align with the target distribution but only aims to find some modes in the distribution. More principled approaches for aligning flow matching models include the recent method of Adjoint Matching [15], which by turning flow matching models into equivalent SDEs computes a gradient matching target for the velocity field with an adjoint ODE. Solving such an ODE is however expensive, especially in cases of foundation models where accurate adjoint ODE solving requires smaller time steps in ODE solver.

**Optimal control and machine learning.** Optimal control (OC) is concerned with steering systems subject to random fluctuations so as to minimize a given cost. OC methods, including the subset of stochastic optimal control (SOC), have been employed in a broad range of areas, including the simulation of rare events in molecular dynamics [24, 25, 30, 80], modeling in finance and economics [21, 47], stochastic filtering and data assimilation [43, 54], tackling nonconvex optimization problems [10], management of power systems and energy markets [3, 50], robotic control [23, 60], analysis of mean-field games [9], optimal transport theory [63, 64], the study of backward stochastic differential equations [8], and large-deviation principles [20]. Relevant and recent applications of SOC in machine learning include performing reward fine-tuning of diffusion and flow matching models [15, 41, 61, 77] and conditional sampling of diffusion processes [13, 48, 66]. There is also a growing literature on SOC methods for sampling from unnormalized densities, as an alternative to MCMC methods [2, 5, 7, 11, 26, 62, 79]. Additionally, there have been a string of methodological works exploring deep learning loss functions for SOC [14, 16, 44].

## 3 Preliminaries

### 3.1 Flow Matching Models

Flow matching models are a class of generative models that are trained to generate samples sequentially following some reference paths. Specifically, one generates samples $x_1$ with a flow matching model by simulating a trajectory from an initial state $x_0 \sim \mathcal{N}(0, I)$ with the dynamics $\dot{x} = v_\theta(x, t)$. The velocity field $v_\theta(x, t)$ is learned with the flow matching loss:

$$\mathcal{L}(\theta) = \mathbb{E}_{x_1 \sim \mathcal{D}, t \sim \text{Uniform}[0,1]} \left\| v_\theta(x_t, t) - u(x_t|x_1) \right\|^2 \tag{1}$$

where $u(x_t|x_1)$ is a reference conditional velocity field. A popular choice is $u(x_t|x_1) = (1 - t)x_t + tx_1$, adopted by a variant called rectified flows [38].

The probability flow $p(x, t)$ corresponding to the velocity field $v(x, t)$ satisfies the so-called continuity equation:

$$\frac{\partial}{\partial t} p(x, t) + \nabla \cdot \big( p(x, t) v(x, t) \big) = 0. \tag{2}$$

Since a flow matching model is modeled as an ordinary differential equation (ODE), one may use any ODE solver to generate samples, including the simplest Euler sampler: $x_{t+\Delta t} = x_t + \Delta t v(x_t, t)$ where $\Delta t$ is the step size.

### 3.2 Optimal Control

In optimal control theory, we aim to find an optimal control signal $u^*(x, t)$ under a known time-varying dynamics $\dot{x} = f(x, u^*, t)$ with the initial state $x(0) = x_0$ such that a cost functional is minimized. The standard control objective is defined as

$$\arg\min_u J[u], \quad J[u] \triangleq \int_0^T L(x(t), u(t), t) \, \mathrm{d}t + \Phi(x(T)), \tag{3}$$

where $\Phi(\cdot)$ is the terminal cost and $L(x, u, t)$ is the running cost.

The Hamilton-Jacobi-Bellman (HJB) equation is indeed the continuous counterpart of the Bellman equation in the discrete domain, where the transitions are defined by a graph instead of a dynamical system $\dot{x} = v(x, t)$. The solution of this optimal control problem satisfies the HJB equation:

$$-\frac{\partial V}{\partial t}(x, t) = \min_u \left[ L(x, u, t) + \nabla V(x, t)^\top f(x, u, t) \right], \tag{4}$$

**Algorithm 1 VGG-Flow** algorithm
___

**Require:** Pretrained flow matching model $v_{\text{base}}(x, t)$, given reward function $r(x_1)$, value gradient model $g_\phi(x, t)$ parameterized by Equation 16.
**Ensure:** Finetuned flow matching model $v_\theta(x, t)$
  Initialize flow matching model $v_\theta \leftarrow v_{\text{base}}$.
  **while** Stopping criterion not met **do**
      Collect trajectories $\{x_t\}_t$ via solving the current neural ODE $\dot{x}_t = v_\theta(x_t, t)$.
      Update value gradient model $g_\phi(x, t)$ with loss $\mathcal{L}_{\text{consistency}}(\phi) + \alpha \mathcal{L}_{\text{boundary}}(\phi)$.
      Update velocity field model $v_\theta(x, t)$ with loss $\mathcal{L}_{\text{matching}}(\theta)$.
  **end while**
___

in which $V(x, t)$ is the value function or the minimal cost-to-go from state $x$ at time $t$:

$$V(x, t) = \min_u \left\{ \int_t^T L(x(s), u(s), s) \ \mathrm{d}s + \Phi(x(T)) \mid x(t) = x \right\}. \tag{5}$$

## 4 Method

### 4.1 Gradient Matching for Aligning Flow Matching Models

Given a reward function $r(\cdot)$, we want to train our generative model to achieve high reward scores for the generated samples and also to preserve the prior distribution of the pretrained model. We can then define the following formulation for training a flow matching model $\mathrm{d}x_t = v_\theta(x_t, t) \, \mathrm{d}t, t \in [0, 1]$ that transforms a standard Gaussian distribution $p_0 = \mathcal{N}(0, I)$ to a target distribution $p_\theta$.

We start with the following optimal control formulation for flow matching alignment problems

$$\min_\theta \mathbb{E}_{x_0 \sim p_0, \dot{x}_t = v_\theta(x_t, t)} \left[ \frac{\lambda}{2} \int_0^1 \|\tilde{v}_\theta(x_t, t)\|^2 \, \mathrm{d}t - r(x_1) \right], \ v_\theta(x_t, t) \triangleq v_{\text{base}}(x_t, t) + \tilde{v}_\theta(x_t, t), \tag{6}$$

where $\tilde{v}_\theta = v_\theta - v_{\text{base}}$ is the residual velocity field and $\lambda$ is the reward multiplier/temperature. With such relationship, we interchangeably use $v_\theta$ and $\tilde{v}_\theta$ to denote the parameterized flow matching model. This program can be interpreted as a control problem where we want to find a deterministic control parameterized by $\tilde{v}$ that minimizes the expected cost of the system, which is defined as the sum of the terminal reward function $r(x_1)$ and the running cost (*i.e.*, regularization term) $\frac{\lambda}{2} \int_0^1 \|\tilde{v}(x_t, t)\|^2 \, \mathrm{d}t$.

**Remark 1** (Connection to Equation 3). In our reward funeting setup, the control $u$ from the general optimal control formulation in Section 3.2 is denoted as $v$ and the terminal time $T$ is set to 1. Our dynamics here is $\dot{x} = f(x, v, t) \triangleq v(x, t)$, the running cost is defined as in $L(x, v, t) \triangleq \frac{\lambda}{2} \|v(x, t) - v_{\text{base}}(x, t)\|^2$, the terminal cost is $\Phi(x(T)) \triangleq -r(x_1), T = 1$, and the value function is $V(x, t) \triangleq \min_v \int_t^1 \frac{\lambda}{2} \|v(x_s, s) - v_{\text{base}}(x_s, s)\|^2 \, \mathrm{d}s - r(x_1)$ for a dynamic starting with $x_t = x$.

The corresponding HJB equation for the above objective is:

$$\partial_t V(x, t) + \min_{\tilde{v}} \left[ \nabla V(x, t) \cdot \left( v_{\text{base}}(x, t) + \tilde{v}(x, t) \right) + \frac{\lambda}{2} \|\tilde{v}(x, t)\|^2 \right] = 0, \tag{7}$$

With the first-order condition of the minimization program of $\tilde{v}$ in the HJB equation, we obtain the following optimal control law:

$$\text{(Value Gradient Matching)} \qquad \tilde{v}^\star(x, t) = -\frac{1}{\lambda} \nabla V(x, t). \tag{8}$$

This optimal control law can be interpreted as a gradient matching criterion, where the residual velocity field $\tilde{v}^\star(x, t)$ should match value function gradient $\nabla V(x, t)$ at state $x$ at time $t$. If an oracle value function is provided, then alignment of the flow matching model can simply be achieved through a "gradient matching" loss between the residual velocity field and the oracle value gradient.

### 4.2 Solving HJB Equation with Value Gradient Guidance

With the optimal control law (Equation 8), the HJB equation reduces to

$$\text{(Value Consistency)} \qquad \frac{\partial}{\partial t} V(x, t) = \frac{1}{2\lambda} \|\nabla V(x, t)\|^2 - \nabla V(x, t) \cdot v_{\text{base}}(x, t). \tag{9}$$

While we could in principle solve this equation by parametrizing $V(x,t)$ with a neural net, it is better that we directly parametrize $\nabla V(x,t)$ since it is considerably more effective and robust, as shown in diffusion model and energy-based model literature [55, 59]. With $g_\phi(x,t) \triangleq \nabla V_\phi(x,t)$, we may write the equivalent gradient-version HJB equation by taking gradients on both sides:

$$\frac{\partial}{\partial t} g_\phi = \frac{1}{\lambda}[\nabla g_\phi]^T g_\phi - [\nabla g_\phi]^T v_{\text{base}}(x,t) - [\nabla v_{\text{base}}(x,t)]^T g_\phi \tag{10}$$

$$= [\nabla g_\phi]^T \left( \frac{1}{\lambda} g_\phi - v_{\text{base}}(x,t) \right) - [\nabla v_{\text{base}}(x,t)]^T g_\phi \tag{11}$$

with the boundary condition $g_\phi(x,1) = -\nabla r(x)$ at terminal time.

With $\beta = 1/\lambda$, we write the following set of losses to update value function gradient model $g_\phi(x,t)$:

$$\mathcal{L}_{\text{consistency}}(\phi) = \mathbb{E}_{x_0 \sim \mathcal{N}(0,I), \dot{x}_t = v(x_t,t)} \left\| \frac{\partial}{\partial t} g_\phi + [\nabla g_\phi]^T (v_{\text{base}} - \beta g_\phi) + [\nabla v_{\text{base}}]^T g_\phi \right\|^2, \tag{12}$$

$$\mathcal{L}_{\text{boundary}}(\phi) = \mathbb{E}_{x_0 \sim \mathcal{N}(0,I), \dot{x}_t = v(x_t,t)} \left\| g_\phi(x_1,1) + \nabla r(x_1) \right\|^2. \tag{13}$$

In practice, this consistency loss based on Equation 9 can be efficiently implemented with finite difference methods and Jacobian-vector products in PyTorch.

Furthermore, with a decently learned value gradient model that captures the optimal control, we regress our residual velocity field to it to learn our flow matching model $v_\theta$:

$$\mathcal{L}_{\text{matching}}(\theta) = \mathbb{E}_{x_0 \sim \mathcal{N}(0,I), \dot{x}_t = v(x_t,t)} \left\| \tilde{v}_\theta(x_t,t) + \beta g_\phi(x_t,t) \right\|^2. \tag{14}$$

Notice that we only use this objective to update $\theta$, not $\phi$. This makes the total training objective $\mathcal{L}_{\text{total}}$

$$\mathcal{L}_{\text{total}}(\theta,\phi) = \mathcal{L}_{\text{matching}}(\theta) + \mathcal{L}_{\text{consistency}}(\phi) + \alpha \mathcal{L}_{\text{boundary}}(\phi), \tag{15}$$

where $\alpha$ is a coefficient to tune the importance of boundary condition loss in the training.

**Efficient parametrization of value function gradients.** Solving the consistency equation for the value function gradient model $g_\phi(x,t)$ can take a non-trivial amount of time. For flow matching models, especially variants like rectified flows, the value of $x_t$ can be well approximated by the reward of the single-Euler-step prediction $\hat{x}_1 = \hat{x}_1(x_t,t) \triangleq x_t + (1-t) \cdot \texttt{stop-gradient}(v(x_t,t))$, in which the stop gradient operation is inspired by DreamFusion [49] and helps improve results. Therefore, we propose to parametrize $g_\phi(x,t)$ with

$$g_\phi(x,t) \triangleq -\eta_t \cdot \texttt{stop-gradient}\left( \nabla_{x_t} r(\hat{x}_1(x_t,t)) \right) + \nu_\phi(x_t,t) \tag{16}$$

where $\eta_t$ is a positive weighting scalar and $\nu_\phi(x_t,t)$ is a learnable error correction term which is supposed to be close to zero when $t \to 1$.

**Putting everything together.** At each training step, our **VGG-Flow** algorithm simulates trajectories $\dot{x}_t = v_\theta(x_t,t)$ with an ODE solver, and use the obtained trajectory data to update the value gradient model $g_\phi$ and velocity field model $v_\theta$. We summarize the proposed method in Algorithm 1.

## 5 Experiments

### 5.1 Experiment Settings

**Base model.** Throughout the paper, we consider the popular open-sourced text-conditioned flow matching model Stable Diffusion 3 [17] and a 20-step Euler solver to sample trajectories.

**Reward model.** We consider three reward models learned from large-scale human preference datasets: Aesthetic Score [33], Human Preference Score (HPSv2) [67, 68], and PickScore [32].

**Prompt dataset.** For Aesthetic Score, we use a set of simple animal prompts used in the original DDPO paper [6]; for HPSv2, we consider photo+painting prompts from the human preference dataset (HPDv2) [67]; for PickScore, we use the prompt set in the Pic-a-Pick dataset [32].

**Metrics.** We follow previous works [15, 41] and compute the variance of latent features (both DreamSim features [22] and CLIP features [29, 53]) extracted from a batch of generated images (we use a batch of size 16) to measure sample diversity. To measure the degree of prior preservation, we

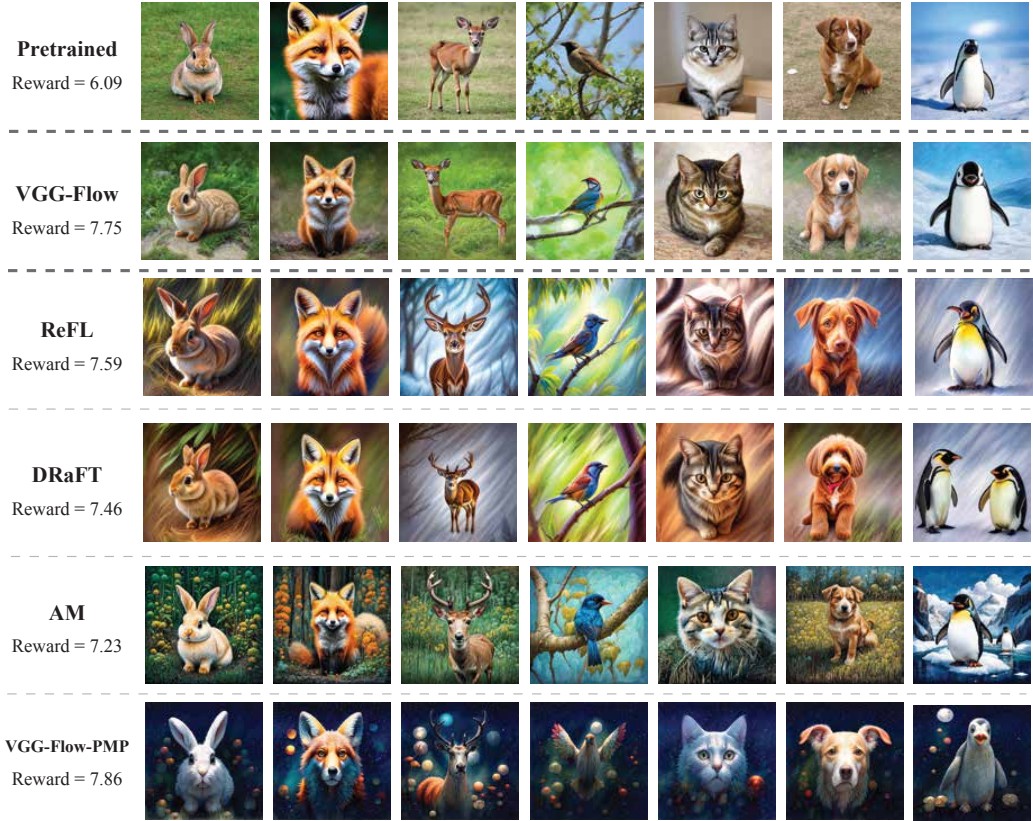

**Pretrained**
Reward = 6.09

**VGG-Flow**
Reward = 7.75

**ReFL**
Reward = 7.59

**DRaFT**
Reward = 7.46

**AM**
Reward = 7.23

**VGG-Flow-PMP**
Reward = 7.86

Figure 2: Comparison on samples generated by models finetuned with different methods. All models are finetuned with a maximum of 400 update steps and for fair qualitative comparison we pick the model checkpoints that yield the best rewards without significant collapsing in image semantics (as ReFL and DRaFT are more prone to overfitting). For each set of images produced by each method, we display their average reward on the left.

| Method | Aesthetic Score | | | HPSv2 | | | PickScore | | |
| | Reward ($\uparrow$) | Diversity DreamSim ($\uparrow$, $10^{-2}$) | FID ($\downarrow$) | Reward ($\uparrow$, $10^{-1}$) | Diversity DreamSim ($\uparrow$, $10^{-2}$) | FID ($\downarrow$) | Reward ($\uparrow$) | Diversity DreamSim ($\uparrow$, $10^{-2}$) | FID ($\downarrow$) |
|---|---|---|---|---|---|---|---|---|---|
| Base (SD3) | $5.99 \pm 0.01$ | $23.12 \pm 0.15$ | $212 \pm 5$ | $2.80 \pm 0.05$ | $22.42 \pm 0.29$ | $558 \pm 2$ | $21.81 \pm 0.02$ | $27.81 \pm 0.10$ | $589 \pm 5$ |
| ReFL | $\mathbf{10.00} \pm 0.31$ | $5.59 \pm 1.33$ | $1338 \pm 191$ | $\mathbf{3.87} \pm 0.01$ | $14.08 \pm 0.55$ | $1195 \pm 21$ | $23.19 \pm 0.05$ | $17.71 \pm 0.77$ | $997 \pm 15$ |
| DRaFT | $9.54 \pm 0.14$ | $7.78 \pm 0.60$ | $1518 \pm 111$ | $3.76 \pm 0.02$ | $15.05 \pm 1.23$ | $1177 \pm 29$ | $23.00 \pm 0.08$ | $19.03 \pm 0.92$ | $\mathbf{968} \pm 26$ |
| AM | $6.87 \pm 0.17$ | $\mathbf{22.34} \pm 2.39$ | $465 \pm 93$ | $3.59 \pm 0.03$ | $14.11 \pm 0.26$ | $1246 \pm 24$ | $22.78 \pm 0.04$ | $19.70 \pm 1.08$ | $1033 \pm 59$ |
| VGG-Flow-PMP | $7.52 \pm 0.16$ | $11.17 \pm 1.67$ | $1170 \pm 213$ | $3.57 \pm 0.05$ | $15.36 \pm 0.06$ | $1195 \pm 5$ | $22.10 \pm 3.61$ | $16.78 \pm 1.13$ | $1148 \pm 47$ |
| **VGG-Flow** | $8.24 \pm 0.07$ | $22.12 \pm 0.17$ | $\mathbf{375} \pm 25$ | $3.86 \pm 0.03$ | $\mathbf{18.40} \pm 1.12$ | $\mathbf{1161} \pm 19$ | $\mathbf{23.21} \pm 0.05$ | $\mathbf{20.93} \pm 0.98$ | $1058 \pm 31$ |

Table 1: Comparison between the models finetuned with our proposed method **VGG-Flow** and the baselines. All models are finetuned with 400 update steps. Since there are inherent trade-offs between reward and other metrics, the values at the final update step do not fully capture the differences between methods. We therefore refer readers to the Pareto front comparisons in Figs. 6, 8, and 10 for a more comprehensive evaluation.

compute the per-prompt FID score between image sets generated by the finetuned model and the base model and use the average per-prompt FID score as the prior preservation metric.

**Baselines.** We consider two types of baselines: the generic ones that rely on direct reward maximization with truncated computation graphs, including ReFL [70] and DRaFT [12], and the adjoint-based Adjoint Matching method [15]. Specifically, ReFL samples a trajectory and take the truncated computational graph stop-gradient$(x_t) \to x_{t+\Delta t}$ with a randomly sampled time step $t$. The model is finetuned to maximize the reward $r(\hat{x}_1(x_t, t))$ of the single-step prediction $\hat{x}_1(x_t, t)$. Similarly, DRaFT truncate the full inference computational graph at some random time step $1 - K\Delta t$ and perform backpropagation on this length $K$ graph with the differentiable reward signal $r(x_1)$. For optimal-control-based baselines, we consider adjoint matching [15], which finetunes flow matching model under stochastic settings (details in Appendix A.1). Additionally, we consider a variant

| Pretrained | VGG-Flow | ReFL | DRaFT | AM | VGG-Flow-PMP |
|:---:|:---:|:---:|:---:|:---:|:---:|

Prompt: Cute cats in conceptual art style.

Prompt: A cat with bunny ears.

Prompt: Footage of an astronaut in a tropical beach.

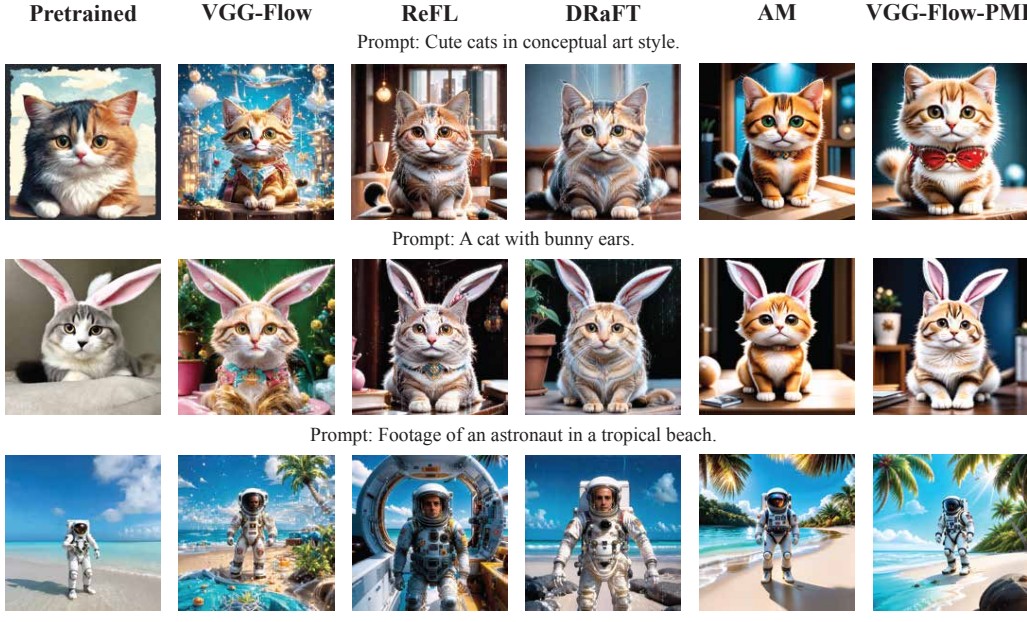

Figure 3: Qualitative results on HPSv2.

| Pretrained | VGG-Flow | ReFL | DRaFT | AM | VGG-Flow-PMP |
|:---:|:---:|:---:|:---:|:---:|:---:|

Prompt: An anthropomorphic white rabbit, male wizard face, dressed in black and white, fine art, award-winning, intricate, elegant, sharp focus, cinematic lighting, highly detailed, digital painting, 8k concept art, art by guweiz and z. w. gu, masterpiece, trending on artstation, 8k.

Prompt: A surfing steampunk giraffe in a rainforest.

Prompt: Flying horses with wings, at sunset at the Lower Galilee.

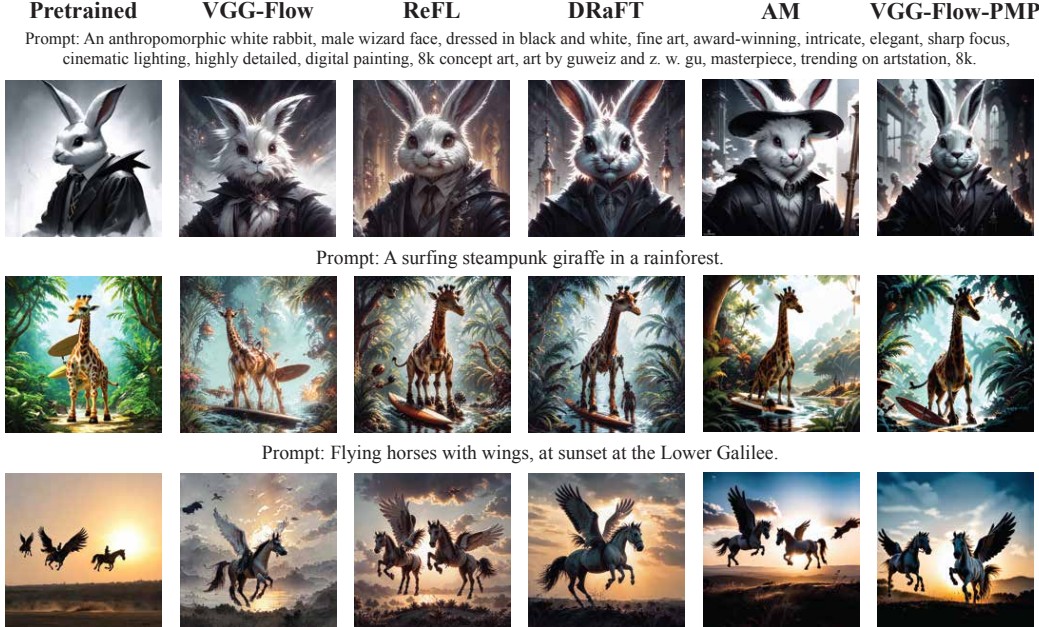

Figure 4: Qualitative results on PickScore.

of VGG-Flow which, derived with the Pontryagin's Maximum Principle [35, PMP], follow an adjoint-matching-like algorithm but with slightly different evolution equations (Appendix A.1).

**Experiment settings and implementation details.** We use LoRA parametrization [31] on attention layers of the finetuned flow matching model with a LoRA rank of 8. The value gradient network in **VGG-Flow** is set to be a scaled-down version of the Stable Diffusion-v1.5 U-Net, initialized with tiny weights in the final output layers. Since Stable Diffusion 3 is a latent flow matching model, the reward for a sampled image $x_1$ is $r(\text{decode}(x_1))$ where $\text{decode}(\cdot)$ is the VAE decoder of the Stable Diffusion 3 model. This decoder is always frozen and we only finetune the LoRA parameters. For all experiments, we use 3 random seeds. For Aesthetic Score, HPSv2 and PickScore experiments, we set the default inverse temperature terms $\beta = 1/\lambda$ to 5e4, 3e7 and 5e5, respectively;

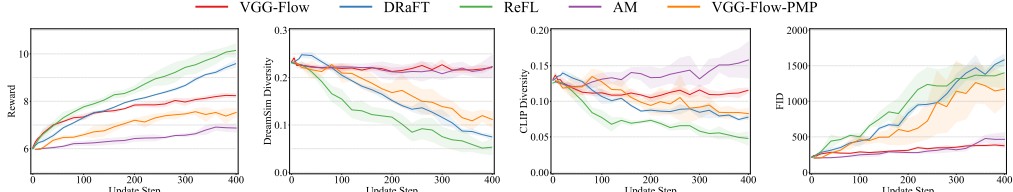

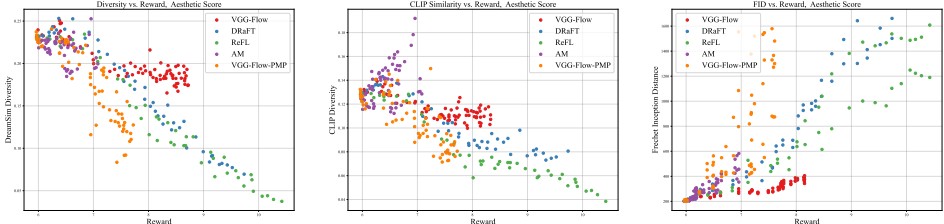

Figure 5: Convergence curves of different metrics for different methods throughout the finetuning process on Aesthetic Score. Finetuning with our proposed **VGG-Flow** converges faster than the non-gradient-informed methods and with better diversity- and prior-preserving capability.

Figure 6: Trade-offs between reward, diversity preservation and prior preservation for different reward finetuning methods on Aesthetic Score. Dots represent the evaluation results of models checkpoint saved after every 5 iterations of finetuning, where ones with greater reward, greater diversity scores and smaller FID scores are considered better.

for all ablation studies with Aesthetic Score, we set $\beta = 1e4$. We set the boundary loss coefficient $\alpha$ to 10000 for all experiments. We use an effective batch size of 32 for all methods, and only use on-policy samples without any replay buffer. For **VGG-Flow**, we sub-sample the collected trajectories by uniformly splitting each into 5 bins and then taking one transition out of each; we also clip the computed reward gradients in Eqn. 16 at the 80th percentile of the gradient norms of the corresponding training batches. For ReFL, we sample the truncation time step between 15 and 20. We follow prior work [51, 70] use $\text{ReLU}(r(x))$ as the reward model for both ReFL and DRaFT for stable training. For experiments on adjoint matching (AM), we use 4 GPUs for each run and set the inverse temperature $\beta$ to $5 \times 10^3, 3 \times 10^5, 1 \times 10^4$ for Aesthetic, HPS, PickScore, respectively, based on the same hyperparameter choosing protocol from [15]. All AM experiments use float32 computation and drop samples that result in too large gradient norms.

## 5.2 Results

**General experiments.** We show in Figure 2 the visualization of samples produced by both the base model and the models finetuned on Aesthetic Score. As ReFL and DRaFT finetuning can easily lead to reward hacking, we perform early stopping and pick the model checkpoints without major loss of image semantics and with the highest reward values possible. Compared to ReFL and DRaFT, out proposed **VGG-Flow** produces higher rewards with better preservation of semantic prior from the base Stable Diffusion 3 model. Our method also works well on other reward models, including HPSv2 and PickScore, as shown in Figure 3 and 4. To further illustrate the advantage of our method and the tradeoffs between reward convergence and other metrics, we present quantitative results in Table 1, Figure 5 and Figure 6. Specifically, we observe that **VGG-Flow** achieves comparable speed with respect to direct reward maximization methods (ReFL and DRaFT) but better maintains sample diversity (measured by DreamSim and CLIP diversity score) and base model prior (measure by FID score). We observe that ReFL and DRaFT on Aesthetic Score easily achieves reward values close to 9, of which value typically indicates complete forgetting of base model prior [12]. Furthermore, the Pareto front figures of reward values, diversity scores and FID scores show that our **VGG-Flow** achieves better diversity/FID scores at the same level of reward values – demonstrating that our **VGG-Flow** outperforms the baselines even if we perform early stopping.

**Effect of reward temperature.** We conduct an ablation study on Aesthetic Score with different $\beta \in \{5000, 10000, 50000\}$ and show in Figure 11 the effect of reward temperature. We observe that for all reward temperatures, the reward smoothly increases at a speed proportional to $\beta$. The sample diversity and prior preservation capability are generally worse with greater $\beta$ values. Greater $\beta$ values also leads to worse trade-off on FID vs. reward but no significant difference in diversity vs. reward.

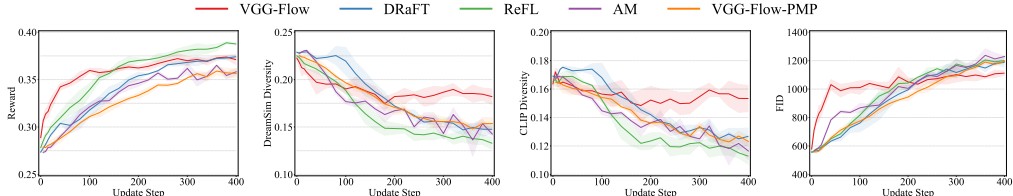

Figure 7: Convergence of different metrics for different methods throughout the finetuning process on HPSv2.

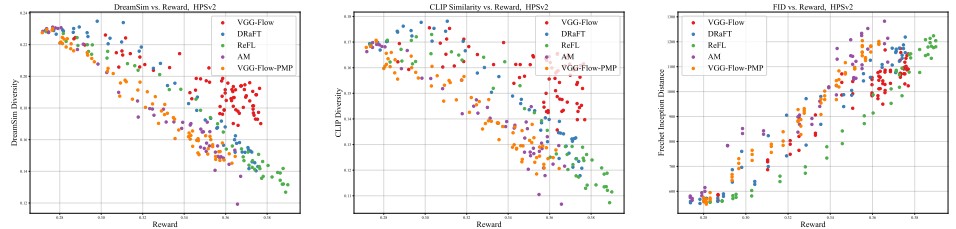

Figure 8: Trade-offs between metrics for different reward finetuning methods (experiments on HPSv2).

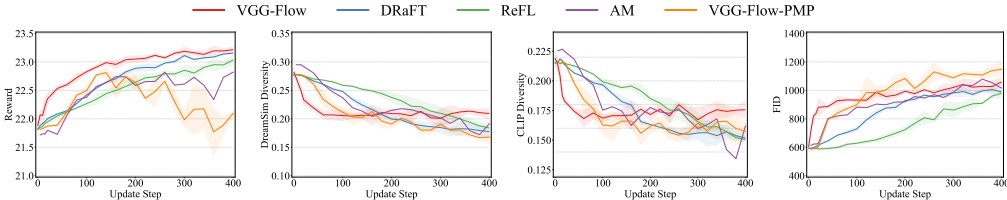

Figure 9: Convergence of different metrics for different methods throughout the finetuning process on PickScore.

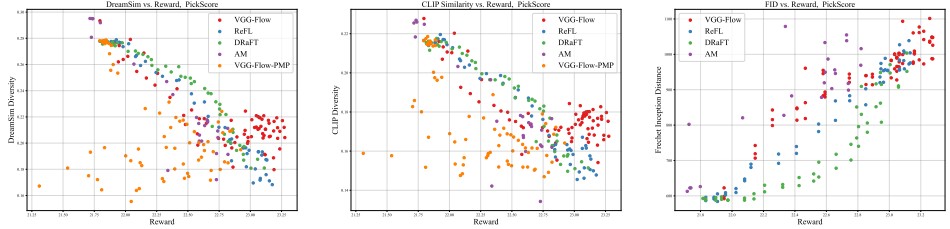

Figure 10: Trade-offs between metrics for different reward finetuning methods (experiments on PickScore).

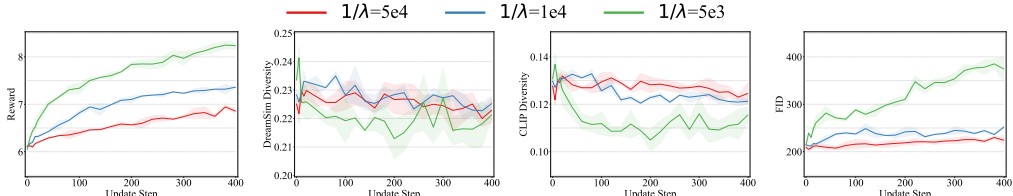

Figure 11: Evolution of metrics for different reward temperature (experiments on Aesthetic Score). Higher temperature $\beta$ leads to faster convergence but with less diversity and less prior preservation.

**Effect of $\eta$ schedule.** We observe that by setting $\eta_t = t$, the convergence speed is faster than our default choice of quadratic schedule $\eta_t = t^2$ (Fig. 13 and 14). Both schedule yields nearly identical trade-offs between metrics, which not only suggest that relative independence of the final performance of trained models on the choice of the parameterization of the learned value gradient model.

**Effect of transition subsampling rate.** We also investigate if a lower subsampling rate, with which the variance of estimated parameter gradients are lower, leads to better performance. In Fig. 15 and 16, we observe that there is no significant difference between subsampling rates of 25% and 50%.

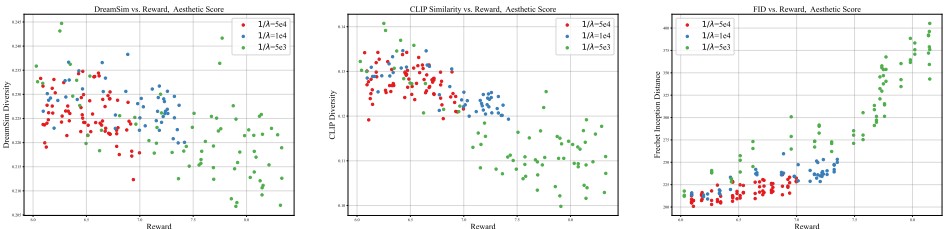

Figure 12: Trade-offs between metrics for different reward temperatures (experiments on Aesthetic Score).

## 6 Discussions

**HJB vs. PMP.** Another way to characterize the optimal control is through Pontryagin's Maximum Principle [35, PMP]. With the control formulation in Section 3.2, we can define the Hamiltonian $H(x, u, t, a) \triangleq L(x, u) + af(x, u, t)$ and the adjoint state (also known as co-state) $a(t)$ that satisfies

$$\dot{a}(t) = -\nabla_x H(x(t), u(t)) \quad \text{s.t.} \quad a(T) = \nabla\Phi(x(T)), \ \dot{x} = f(x, u, t). \tag{17}$$

Essentially, the PMP states that the optimal control $u^*$ satisfies $u^*(t) = \arg\max_u H(x^*(t), u, t, a^*(t))$, where $x^*$ and $a^*$ are the solutions to equation 17 for the optimal control $u^*$. With the cost functional and dynamics in our setting (Equation 3), we have $\dot{a} = -2\nabla[\|v - v_{\text{base}}\|^2 + v^T a]$ and $\tilde{v}_\theta(x, t) + a(t) = 0$. By comparing it with Equation 8, we have $a(t) = \nabla V(x_t, t)$ for any trajectory $x_{t \in [0,1]}$ from the dynamics $\dot{x} = v(x, t)$. While mathematically equivalent, solving this adjoint equation not only requires the expensive (and often unaffordable) computation of $\nabla H$ multiple times per trajectory but also is prone to accumulated errors in solving the adjoint equation. In contrast, our HJB-based method is more efficient and robust because it 1) solves for $\nabla V$ in an amortized approach with the forward-looking parametrization of $\nabla V$ and 2) allows for efficient transition subsampling. See Section A.1 for more details.

**Connection with adjoint matching.** Adjoint matching [15] reaches a matching objective similar to the one in the above PMP discussion. However, their framework is based on stochastic optimal control instead of deterministic optimal control. While the stochastic setting allows them to sample from a simple tilted distribution $p_{\text{base}}(x) \exp r(x)$, their algorithm requires modifying the flow matching ODE into an SDE with equal marginals. Our proposed algorithm fine-tunes directly the ODE dynamics with deterministic control. Computationally, adjoint matching relies on solving the adjoint ODE, which requires taking one backward pass through the model for each time step. VGG-Flow relies on the value function gradient model of current step and is thus more computationally tractable.

**Limitations.** Since our method relies on a relaxed objective, the finetuned distribution approximates the ideal KL-regularized distribution well only in the case of a relatively small $\lambda$. Implementation-wise, we use finite differences to approximate the first-order gradients of the value gradient estimator and disable all second-order gradients during backpropagation, which inevitably leads to biases. Furthermore, our method suffers from the same challenge of exploration-exploitation tradeoff as in common reinforcement learning settings. As we aim for fast convergence within limited computational resources, our hyperparameter settings are in theory more prone to mode collapse. Furthermore, we do not explore better architecture designs, which is shown important for efficient and stable finetuning of foundation models [37, 52].

## 7 Conclusion

We propose **VGG-Flow**, an efficient and robust method for performing alignment of flow matching models with some reward model. By leveraging a relaxed objective and the HJB equation in optimal control theory, we derive a gradient matching method that allows us to finetune flow matching models with probabilistic guarantees and memory-efficient computation. We empirically demonstrate the effectiveness of our **VGG-Flow** on Stable Diffusion 3, a popular large-scale text-conditioned flow matching model, with common image-input reward functions. As for broader impact, we point out that improving the alignment of flow matching models enhances their ability to reliably follow human instructions, contributing to the development of more trustworthy and controllable AI systems that can better serve societal needs in education, healthcare, and decision support.

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

# Appendix

## Table of Contents

# A   Theoretical Connections between Optimal Control Formulations

## A.1   Deriving adjoint matching from Pontryagin's maximum principle

For a general dynamical system $\dot{x} = f(x, u, t)$ and control problem $\min_u \int_0^T L(x(t), u(t), t)\, \mathrm{d}t + \Phi(x(T))$, we define the Hamiltonian $H(x, u, t, a) \triangleq L(x, u) + a f(x, u, t)$ and the adjoint state $a(t)$ that satisfies $\dot{a}(t) = -\nabla H$ and $a(T) = \nabla \Phi(x(T))$. The Pontryagin's maximum principle (PMP) states that the optimal control $u^*$ satisfies $u^*(t) = \arg\max_u H(x^*(t), u, t, a^*(t))$.

For our flow matching model $\dot{x}_t = v_\theta(x_t, t)$ setup and the control formulation in Remark 1, we have

$$\min_\theta \mathbb{E}_{\dot{x}_t = v_\theta(x_t, t)} \left[ \frac{\lambda}{2} \int_0^1 \|v_\theta(x_t, t) - v_{\text{base}}(x_t, t)\|^2 - r(x_1) \right] \mathrm{d}t \tag{18}$$

$$H(x, v_\theta, t, a) = \frac{\lambda}{2} \|v_\theta(x, t) - v_{\text{base}}(x, t)\|^2 + a^\top v_\theta(x, t). \tag{19}$$

$$\dot{a}(x, t) = -\frac{\lambda}{2} \nabla_x \left( \|v_\theta(x, t) - v_{\text{base}}(x, t)\|^2 + a(x, t)^\top v_\theta(x, t) \right), \tag{20}$$

with the terminal constraint $a(x_1, 1) = -\nabla r(x_1)$. In order to solve $v^* = \arg\max_v H(x, v, t, a)$, with such quadratic form we have

$$v^* = v_{\text{base}} - \frac{a}{\lambda} \Rightarrow \tilde{v} + \frac{a}{\lambda} = 0. \tag{21}$$

Therefore, we can have an intuitive algorithm following the common practice of solving PMP:

1. Solve forward ODE, $\dot{x}_t = v_\theta(x_t, t), x_0 \sim \mathcal{N}(0, I)$, to obtain $\{x_t\}_{t=0}^1$;
2. Solve backward ODE in Equation 20, $\dot{a} = -\nabla H, a(x_1, 1) = -\nabla r(x_1)$, to obtain $\{a_t\}_{t=0}^1$;
3. Update velocity field $v_\theta$ by minimizing matching loss

$$\mathcal{L}(\theta) = \int \|\lambda \tilde{v}_\theta(x_t, t) + a_t\|^2 \, \mathrm{d}t. \tag{22}$$

This is already very similar to the adjoint matching algorithm proposed in [15] (but for stochastic control settings), where the authors obtain their algorithm from a different derivation and slightly different assumptions.

Comparing this Equation 21 with Equation 8, we can easily get

$$a(x, t) = \nabla V(x, t), \tag{23}$$

which, maybe not that surprisingly, indicates adjoint matching and **VGG-Flow** share the same vector matching objective (Equation 22 and Equation 8). They differ in the way of how to obtain the matching target (either $a_t$ or $\nabla V$).

## A.2   The adjoint method vs. adjoint matching for deterministic optimal control

Here we give a more general discussion between the adjoint method (for open-loop control) and adjoint matching (for closed-loop control) in the deterministic control setting.

### A.2.1   Open-loop vs closed-loop formulations of deterministic optimal control

Consider the deterministic control problem in equation 3 with a quadratic cost and affine control. That is, $L(x, u, t) = \frac{1}{2} \|u\|_{Q(t)}^2 + f(x, t)$, where $Q(t) \in \mathbb{R}^{d \times d}$ is a positive-definite matrix, and $\|u\|_Q^2 = u^\top Q u$, and the drift is $f_{\text{drift}}(x, u, t) = b(x, t) + u$. Hence, the control problem reads

$$\min_{u:[0,T] \to \mathbb{R}^d} J[u] \triangleq \int_0^T \left( \frac{1}{2} \|u(t)\|_{Q(t)}^2 + f(X_t, t) \right) \mathrm{d}t + \Phi(X_T), \tag{24}$$

$$\text{s.t.} \quad \dot{X}_t = b(X_t, t) + u(t), \qquad X_0 \sim x_0. \tag{25}$$

This is an *open-loop control problem*, because the control $u(t)$ does not depend explicitly on the state $X_t$. Note however that since both the starting point and the dynamics are deterministic, given the

function $u : [0, t] \to \mathbb{R}^d$, it is possible to determine the state $X_t$, and hence $u(t)$ can be defined to depend implicitly on $X_t$.

Alternatively, consider the control problem

$$\min_{u:\mathbb{R}^d \times [0,T] \to \mathbb{R}^d} \tilde{J}[u] \triangleq \int_0^T \left( \frac{1}{2} \|u(X_t, t)\|_{Q(t)}^2 + f(X_t, t) \right) \mathrm{d}t + \Phi(X_T), \tag{26}$$

$$\text{s.t.} \quad \dot{X}_t = b(X_t, t) + u(X_t, t), \qquad X_0 \sim x_0. \tag{27}$$

In this case, $u$ is a function that depends explicitly on the state $X_t$, which makes this a *closed-loop control problem*.

In general, closed-loop control problems are more general than open-loop problems, but in our case both problems are actually equivalent because the deterministic dynamics and initial conditional make it possible to make $u(t)$ depend on $X_t$ implicitly. In other words, for any open-loop control $u : [0, T] \to \mathbb{R}^d$, we can define a closed-loop control $\tilde{u} : \mathbb{R}^d \times [0, T] \to \mathbb{R}^d$ by setting $\tilde{u}(x, t) = u(t)$. And for any closed-loop control $\tilde{u} : \mathbb{R}^d \times [0, T] \to \mathbb{R}^d$, we can define an open-loop control $u : [0, T] \to \mathbb{R}^d$ by setting $u(t) = u(X_t, t)$, where $X = (X_s)_{s \in [0,t]}$ satisfies the ODE in equation 27.

Thus, finding the solution to the open-loop problem in equations 24-25 is equivalent to finding the solution to the closed-loop problem in equations 26-27. As we see next, the former formulation naturally gives rise to the adjoint matching loss for deterministic optimal control, while the latter yields the basic adjoint matching loss, which is simply a reformulation of the adjoint method.

### A.2.2 Solving the closed-loop problem: the adjoint method for deterministic optimal control

Using an argument similar to the one used in [15, Prop. 2] for stochastic optimal control, we can derive the continuous-time version of the basic adjoint matching loss for deterministic control:

**Proposition 2** (Basic adjoint matching for deterministic control). *Consider the adjoint ODE*

$$\frac{\mathrm{d}}{\mathrm{d}t} a(t; X, u) = -\left[ (\nabla_{X_t}(b(X_t, t) + u(X_t, t)))^\top a(t; X, u) + \nabla_{X_t} \left( f(X_t, t) + \frac{1}{2} \|u(X_t, t)\|^2 \right) \right], \tag{28}$$

$$a(1; X, u) = \nabla \Phi(X_1). \tag{29}$$

*Suppose that the control $u : \mathbb{R}^d \times [0, T] \to \mathbb{R}^d$ is parameterized by $\theta$, and let $\tilde{J}[u]$ be the closed-loop control objective in equation 26. Then, the gradient $\nabla_\theta \tilde{J}[u]$ is equal to the gradient of this loss:*

$$\mathcal{L}_{\text{Basic-Adj-Match}}(u) := \frac{1}{2} \int_0^1 \left\| u(X_t, t) + Q(t)^{-1/2} a(t; X, \bar{u}) \right\|^2 \mathrm{d}t, \tag{30}$$

$$X \text{ s.t. } \dot{X}_t = b(X_t, t) + \bar{u}(X_t, t), \quad \bar{u} = \texttt{stop-gradient}(u),$$

*where $\bar{u} = \texttt{stop-gradient}(u)$ means that the gradients of $\bar{u}$ with respect to the parameters $\theta$ of the control $u$ are artificially set to zero.*

*Proof.* The proof mirrors the proof of [15, Prop. 2]. If we define the adjoint state

$$a(t, X, u) = \nabla_{X_t} \left( \int_0^T \left( \frac{1}{2} \|u(X_t, t)\|_{Q(t)}^2 + f(X_t, t) \right) \mathrm{d}t + \Phi(X_T) \right), \tag{31}$$

$$\text{where } X \text{ is a solution of } \dot{X}_t = b(X_t, t) + u(X_t, t), \tag{32}$$

we have that $a(t, X, u)$ satisfies the adjoint ODE in equations 28-29. In analogy with equation 32 of [15], we have that

$$\frac{\mathrm{d}}{\mathrm{d}\theta} \tilde{J}[u] = \frac{1}{2} \int_0^T \frac{\partial}{\partial \theta} \|u(X_t, t)\|_{Q(t)}^2 \mathrm{d}t + \int_0^T \frac{\partial u(X_t, t)}{\partial \theta}^\top a(t, X, u) \mathrm{d}t, \tag{33}$$

where $\frac{\mathrm{d}}{\mathrm{d}\theta}$ and $\frac{\partial}{\partial\theta}$ denote the total and partial derivatives with respect to $\theta$. Completing the square, we have that

$$
\begin{aligned}
&\frac{1}{2}\frac{\partial}{\partial\theta}\|u(X_t,t)\|^2_{Q(t)} + \frac{\partial u(X_t,t)}{\partial\theta}^\top a(t,X,u) \\
&= \frac{1}{2}\frac{\partial}{\partial\theta}\left(u(X_t,t)^\top Q(t)^{1/2}Q(t)^{1/2}u(X_t,t)\right) + \frac{\partial u(X_t,t)}{\partial\theta}^\top Q(t)^{1/2}Q(t)^{-1/2}a(t,X,\bar u) \\
&= \frac{1}{2}\frac{\partial}{\partial\theta}\|u(X_t,t) + Q(t)^{-1/2}a(t,X,u)\|^2_{Q(t)}
\end{aligned} \tag{34}
$$

where $Q(t)^{1/2}$ is defined as the matrix with the same eigenvectors as $Q(t)$ and eigenvalues equal to the square root of the eigenvalues of $Q(t)$, and $\bar u = \texttt{stopgrad}(u)$. Notice adjoint $a$ does not depend on $\theta$. Plugging equation 34 into equation 33, we can finally rewrite the gradient as the gradient of $\mathcal{L}_{\mathrm{Basic-Adj-Match}}$. $\qquad\square$

### A.2.3 Solving the open-loop problem: the adjoint matching loss for deterministic optimal control

**Proposition 3** (Adjoint matching for deterministic control). *Consider the lean adjoint ODE:*

$$
\frac{\mathrm{d}}{\mathrm{d}t}\tilde a(t;X) = -\left[(\nabla_{X_t}b(X_t,t))^\top \tilde a(t;X) + \nabla_{X_t}f(X_t,t)\right], \tag{35}
$$

$$
\tilde a(1;X) = \nabla\Phi(X_1). \tag{36}
$$

*Suppose that the control $u : [0,T] \to \mathbb{R}^d$ is parameterized by $\theta$, and let $J[u]$ be the open-loop control objective in equation 26. Then, the gradient $\nabla_\theta J[u]$ is equal to the gradient of this loss:*

$$
\mathcal{L}_{\mathrm{Adj-Match}}(u) := \frac{1}{2}\int_0^1 \left\|u(t) + Q(t)^{-1/2}\tilde a(t;X)\right\|^2 \mathrm{d}t, \tag{37}
$$

$$
X \text{ s.t. } \dot X_t = b(X_t,t) + \bar u(t), \quad \bar u = \texttt{stop-gradient}(u),
$$

*where $\bar u = \texttt{stop-gradient}(u)$ means that the gradients of $\bar u$ with respect to the parameters $\theta$ of the control $u$ are artificially set to zero.*

*Proof.* The proof mirrors the proof of [15, Prop. 2], and the proof of our Prop. 2. If we define the adjoint state

$$
a(t,X) = \nabla_{X_t}\left(\int_0^T \left(\frac{1}{2}\|u(t)\|^2_{Q(t)} + f(X_t,t)\right)\mathrm{d}t + \Phi(X_T)\right), \tag{38}
$$

$$
\text{where } X \text{ is a solution of } \dot X_t = b(X_t,t) + u(t), \tag{39}
$$

we have that $a(t,X)$ satisfies the adjoint ODE in equations 28-29. Note that unlike in Prop. 2, the control $u(t)$ does not depend on the state $X_t$, which simplifies expressions substantially as $\nabla_{X_t}u(t) = 0$. In analogy with equation 32 of [15] and our equation 33, we have that

$$
\frac{\mathrm{d}}{\mathrm{d}\theta}J[u] = \frac{1}{2}\int_0^T \frac{\partial}{\partial\theta}\|u(t)\|^2_{Q(t)}\,\mathrm{d}t + \int_0^T \frac{\partial u(t)}{\partial\theta}^\top a(t,X)\,\mathrm{d}t, \tag{40}
$$

and completing the square as in equation 34, we obtain that $\frac{1}{2}\frac{\partial}{\partial\theta}\|u(t)\|^2_{Q(t)} + \frac{\partial u(t)}{\partial\theta}^\top a(t,X) = \frac{1}{2}\frac{\partial}{\partial\theta}\|u(t) + Q(t)^{-1/2}a(t,X)\|^2_{Q(t)}$. Plugging this equality into equation 40 concludes the proof. $\quad\square$

# B    Bounds of Resulted Distributions

## B.1    Bounding the Wasserstein-2 Distance

We first analyze the relationship between our objective of **VGG-Flow** and the 2-Wasserstein distance ($W_2$). We show that our objective minimizes a strong upper bound on $W_2(p_1, q_1)$.

Let $p_0 = q_0$ be the initial distribution. Consider the two flows, coupled by their initial condition:

$$\dot{x}_t = v_\theta(x_t, t), \quad x_0 \sim p_0 \implies x_t \sim p_t$$
$$\dot{y}_t = v_{\text{base}}(y_t, t), \quad y_0 = x_0 \sim q_0 \implies y_t \sim q_t$$

By definition, the squared $W_2$ distance is the minimum expected squared distance over all possible couplings. Our choice of $x_0 = y_0$ is one such coupling, so it provides an upper bound:

$$W_2(p_t, q_t)^2 \leq \mathbb{E}[\|x_t - y_t\|^2]$$

**Proposition 4** ($W_2$ Bound via Grönwall's Inequality)**.** *Assume the base vector field $v_{base}$ is $L$-Lipschitz in $x$. Then the $W_2$ distance is bounded by the $L_2$ FM loss:*

$$W_2(p_1, q_1)^2 \leq C \int_0^1 \mathbb{E}_{p_t}[\|\tilde{v}_\theta(x_t, t)\|^2] \, \mathrm{d}t$$

*where $C = e^{2L+1}$ is a constant.*

*Proof.* Let $\Delta_t = x_t - y_t$ and $u(t) = \mathbb{E}[\|\Delta_t\|^2]$. We have $u(0) = \mathbb{E}[\|x_0 - y_0\|^2] = 0$. The time derivative is $\dot{\Delta}_t = v_\theta(x_t, t) - v_{\text{base}}(y_t, t)$. Let $\tilde{v}_\theta(x, t) = v_\theta(x, t) - v_{\text{base}}(x, t)$.

$$\frac{d}{dt}\|\Delta_t\|^2 = 2\langle \Delta_t, \dot{\Delta}_t \rangle$$
$$= 2\langle \Delta_t, [v_\theta(x_t, t) - v_{\text{base}}(x_t, t)] + [v_{\text{base}}(x_t, t) - v_{\text{base}}(y_t, t)] \rangle$$
$$= 2\langle \Delta_t, \tilde{v}_\theta(x_t, t) \rangle + 2\langle \Delta_t, v_{\text{base}}(x_t, t) - v_{\text{base}}(y_t, t) \rangle$$

We apply the Cauchy-Schwarz inequality to the first term and the $L$-Lipschitz condition to the second:

$$\frac{d}{dt}\|\Delta_t\|^2 \leq 2\|\Delta_t\|\|\tilde{v}_\theta(x_t, t)\| + 2\|\Delta_t\|(L\|\Delta_t\|)$$

Using Young's inequality ($2ab \leq a^2 + b^2$) on the first term gives:

$$\frac{d}{dt}\|\Delta_t\|^2 \leq (\|\Delta_t\|^2 + \|\tilde{v}_\theta(x_t, t)\|^2) + 2L\|\Delta_t\|^2 = (2L + 1)\|\Delta_t\|^2 + \|\tilde{v}_\theta(x_t, t)\|^2$$

Taking the expectation and letting $b(t) = \mathbb{E}_{p_t}[\|\tilde{v}_\theta(x_t, t)\|^2]$, we have the differential inequality:

$$\dot{u}(t) \leq (2L + 1)u(t) + b(t)$$

By the integral form of Grönwall's inequality, which states that if

$$\dot{u}(t) \leq au(t) + b(t) \quad \text{with} \quad u(0) = 0, \quad \text{then} \quad u(t) \leq \int_0^t e^{a(t-s)} b(s) ds$$

Applying this with $a = (2L + 1)$, the solution at $t = 1$ is:

$$u(1) \leq \int_0^1 e^{(2L+1)(1-s)} b(s) ds \leq e^{2L+1} \int_0^1 b(s) ds$$

Since $W_2(p_1, q_1)^2 \leq u(1)$, we arrive at the bound:

$$W_2(p_1, q_1)^2 \leq e^{2L+1} \int_0^1 \mathbb{E}_{p_t}[\|\tilde{v}_\theta(x_t, t)\|^2] \, \mathrm{d}t \tag{41}$$

$$\square$$

This result confirms that our objective is a theoretically sound one for minimizing an upper bound on the $W_2$ distance.

## B.2 Bounding the KL Divergence

We now analyze the KL divergence. Unlike the $W_2$ distance, the KL divergence is sensitive to changes in density, which are governed by the *divergence* of the vector field.

The marginal densities satisfy the continuity equations ($t \in (0, 1)$):

$$\partial_t p_t(x, t) = -\nabla \cdot (p_t(x, t) v_\theta(x, t))$$
$$\partial_t q_t(x, t) = -\nabla \cdot (q_t(x, t) v_{\text{base}}(x, t))$$

**Proposition 5** (KL Divergence Identity for ODEs). *Assume the vector fields $v_\theta, v_{base}$ and densities $p_t, q_t$ are sufficiently smooth and have sufficient decay at infinity such that all boundary terms from integration by parts vanish. Then, the exact identity for the final KL divergence is:*

$$\mathcal{D}_{\text{KL}} (p_1 \| q_1) = - \int_0^1 \mathbb{E}_{p_t} \left[ \tilde{v}_\theta(x_t, t) \cdot \nabla \log q_t \right] \mathrm{d}t - \int_0^1 \mathbb{E}_{p_t} \left[ \nabla \cdot \tilde{v}_\theta(x_t, t) \right] \mathrm{d}t,$$

*where $\tilde{v}_\theta = v_\theta - v_{base}$.*

Applying a bound to the first term (as we did for the $W_2$ proof) gives the final inequality:

$$\mathcal{D}_{\text{KL}} (p_1 \| q_1) \leq \underbrace{\frac{1}{2} \int_0^1 \mathbb{E}_{p_t}[\|\tilde{v}_\theta(x_t, t)\|^2] \, \mathrm{d}t}_{\text{(A) } L_2 \text{ Value Gradient Matching Loss}} + \underbrace{C(p, q)}_{\text{(B) Path-Dependent Term}} \underbrace{- \int_0^1 \mathbb{E}_{p_t}[\nabla \cdot \tilde{v}_\theta(x_t, t)] \, \mathrm{d}t}_{\text{(C) Divergence Term}} \quad (42)$$

where $C(p, q) = \frac{1}{2} \int_0^1 \mathbb{E}_{p_t}[\|\nabla \log q_t\|^2] \, \mathrm{d}t$ is a functional that depends on both the target path $q_t$ and the learned path $p_t$.

**Remark 6** (Justification for the $L_2$ Proxy Objective). Equation 42 shows that the KL divergence is bounded by the $L_2$ value gradient matching loss (Term A), a path-dependent term (Term B), and a divergence-dependent term (Term C). Term (B) depends on both the learned path $p_t$ and the target path $q_t$. It can be bounded, for example, if the target score function has a uniform bound (i.e., $\|\nabla \log q_t(x)\| \leq M_t$ for all $x, t$), which would imply $\mathbb{E}_{p_t}[\|\nabla \log q_t\|^2] \leq M_t^2$. The primary challenge is that Term (A) and Term (C) are geometrically independent, and Term (C) is computationally expensive to estimate. We therefore use the value gradient matching loss as a computationally efficient proxy objective. We empirically justify this choice, as our finetuned models produce high-quality samples. This success suggests that for our network architecture and problem setup, minimizing Term (A) is sufficient, and the "missing" divergence term (Term C) is implicitly regularized or remains small, likely due to the implicit bias of the neural network.

## C Experiment Details

### C.1 Finite difference for value consistency

Our value consistency loss requires the costly computation of second-order gradients during back-propagation. To save memory and time, we instead use finite differences to approximate the terms (with $u = v_{\text{base}} - \frac{1}{\lambda} g_\phi$):

$$\frac{\partial}{\partial t} g_\phi(x_t, t) \approx \frac{g_\phi(x_t + \epsilon v(x_t, t)\cdot, t + \epsilon) - g_\phi(x_t, t)}{\epsilon} \tag{43}$$

$$\left( [\nabla g_\phi]^T \left( v_{\text{base}} - \frac{1}{\lambda} g_\phi \right) \right)_{(x_t, t)} \approx \frac{g_\phi \left( x_t + \epsilon \overline{\nabla}[v(x_t, t)], t \right) - g_\phi \left( x_t - \epsilon \overline{\nabla}[v(x_t, t)], t \right)}{2\epsilon} \tag{44}$$

$$\left( [\nabla v_{\text{base}}]^T g_\phi \right)_{(x_t, t)} \approx \frac{v_{\text{base}} \left( x_t + \epsilon \overline{\nabla}[g_\phi(x_t, t)], t \right) - v_{\text{base}} \left( x_t - \epsilon \overline{\nabla}[g_\phi(x_t, t)], t \right)}{2\epsilon} \tag{45}$$

where $\overline{\nabla}(\cdot) = \texttt{stop-gradient}(\cdot)$. The stop gradient operations on nested function calls prevent second-order gradients during backpropagation. Empirically, we find this approximation works well.

### C.2 More implementation details

In our experiments, we choose $\eta_t = t^2$ in Equation 16 if not otherwise specified. We use a CFG scale of $w_{\text{CFG}} = 5.0$ for all experiments, and the velocity fields of both the base and finetuned models are CFG-composited as $v(x, t; c) = (1 + w_{\text{CFG}})v(x, t; c) - w_{\text{CFG}}v(x, t; \varnothing)$. We stop the gradients on $v(x, t; \varnothing)$ as we found this leads to faster convergence. We use the best learning rates (in terms of fast yet stable reward convergence) for each method instead of a fixed ones, as we observe that methods like ReFL and DRaFT can be unstable for very large learning rates. Specifically, we use $5e-4$ for **VGG-Flow** on all reward models, $5e-5$ for **VGG-Flow**-PMP on HPSv2 and PickScore, and $1e-4$ for all others. We use the standard AdamW optimizer with $\beta_1 = 0.9$, $\beta_2 = 0.999$ and weight decay $1e-2$. We clip the norm of network update gradients to $1$. We use bfloat16 computation for the flow matching model but float32 for the reward model due to numerical precision issues.

# D Additional Figures

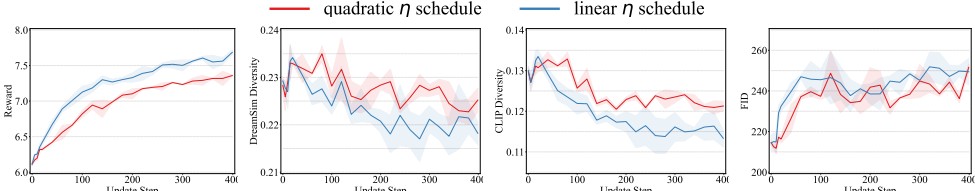

Figure 13: Evolution of metrics for different $\eta$ schedule (experiments on Aesthetic Score). The linear schedule of $\eta$ leads to faster convergence.

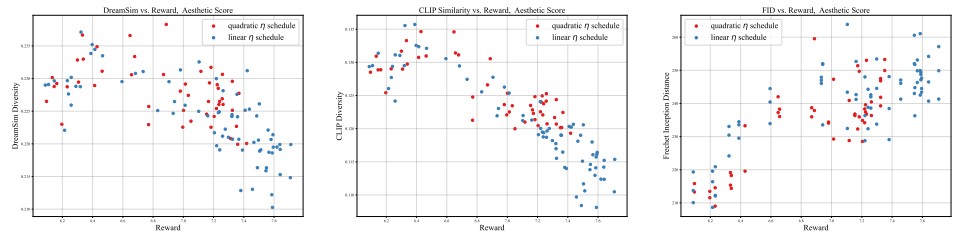

Figure 14: Trade-offs between metrics for different $\eta$ schedule (experiments on Aesthetic Score).

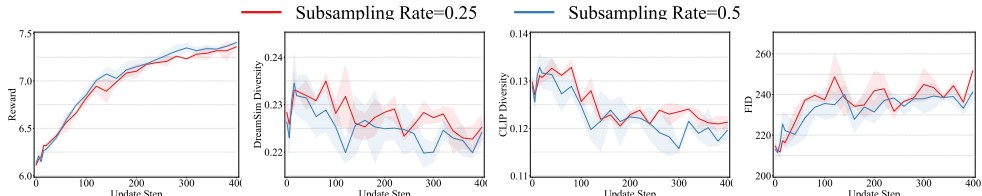

Figure 15: Evolution of metrics for different transition subsampling rates (experiments on Aesthetic Score).

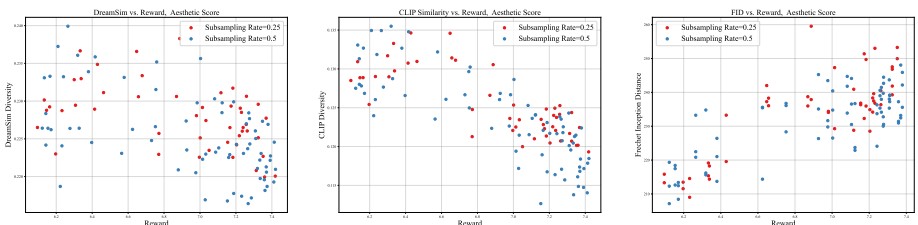

Figure 16: Trade-offs between metrics for different transition subsampling rates (experiments on Aesthetic Score).

# E    Evolution of Generated Samples

We show in Figure 17 that our method is more capable of preserving the prior from the base model during the finetuning process.

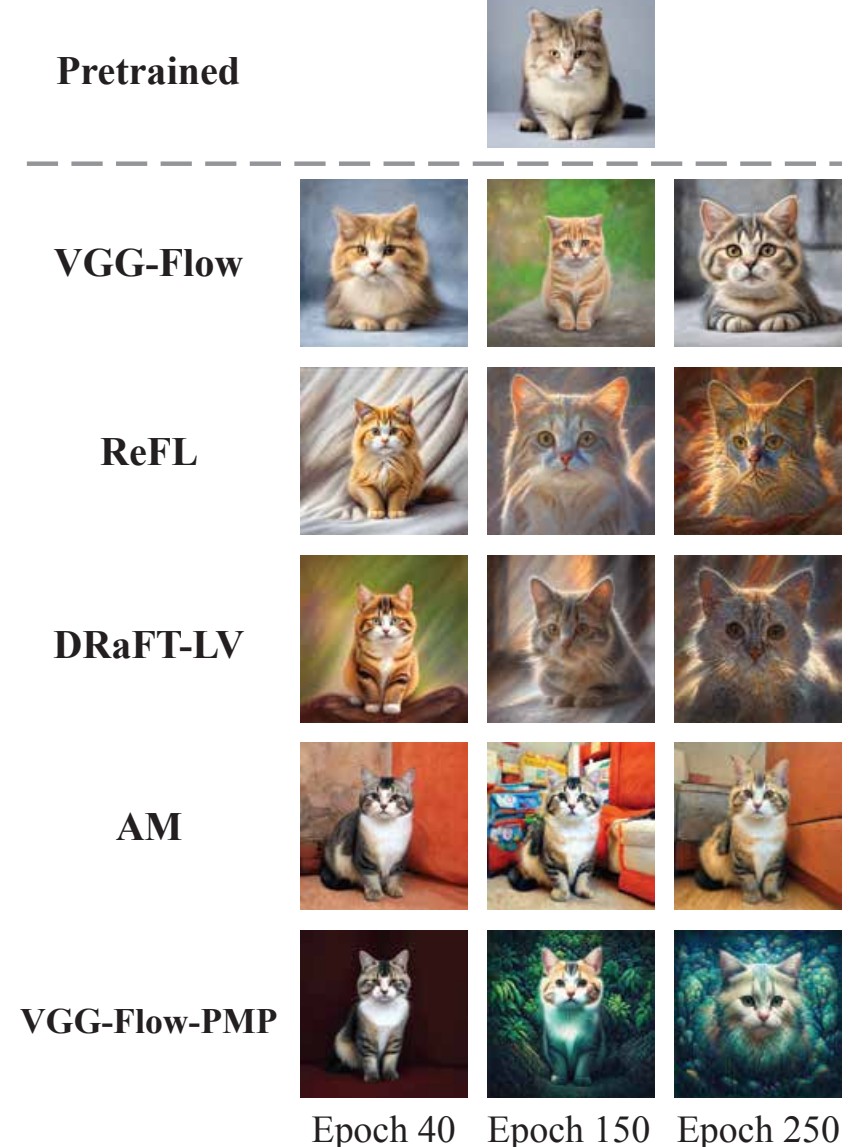

Figure 17: The degradation of image quality of baselines, compared to the evolution sequence of results produced by our method.

# F   More Generated Samples

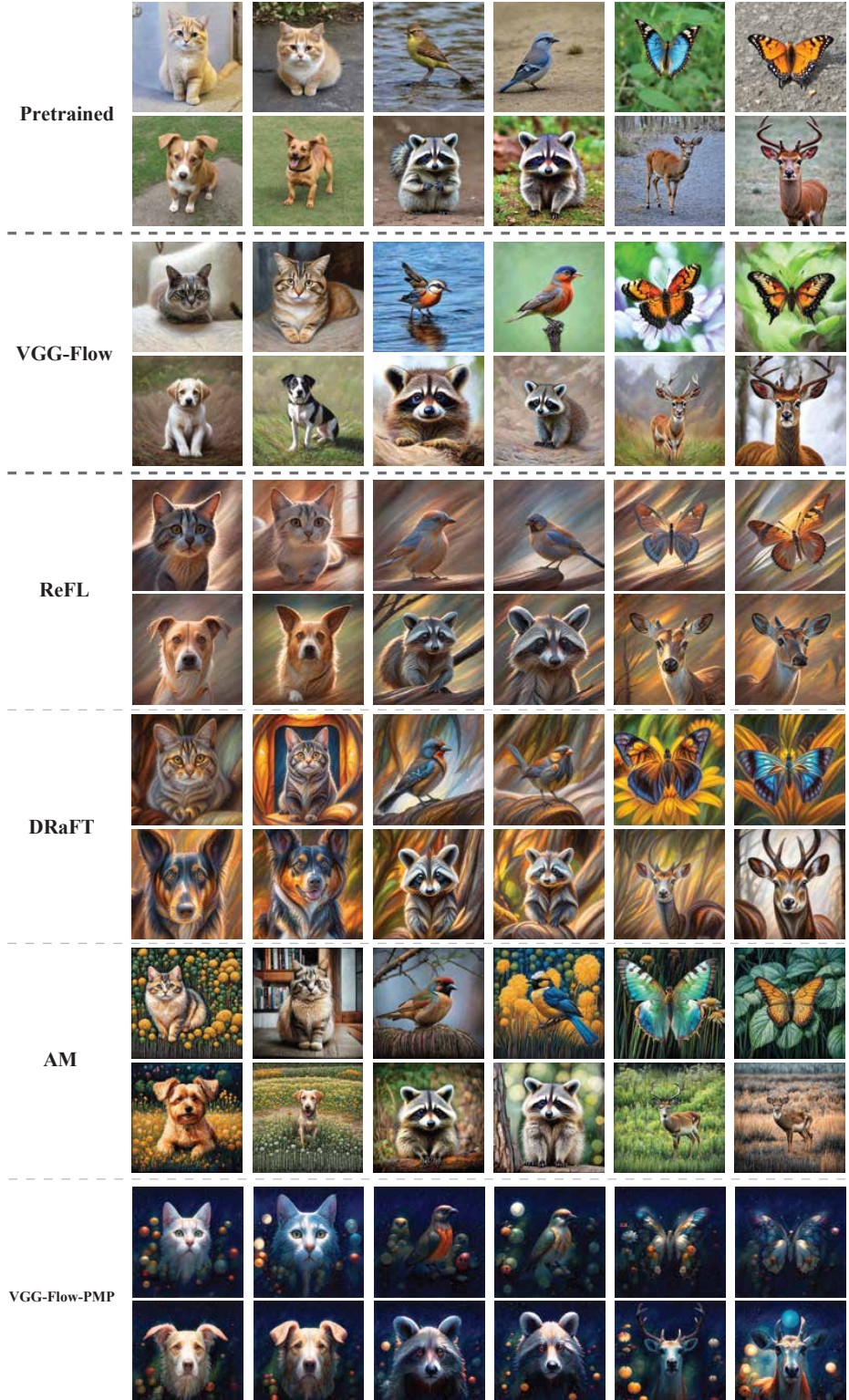

Figure 18: More qualitative results on Aesthetic Score.

| Pretrained | VGG-Flow | ReFL | DRaFT | AM | VGG-Flow-PMP |
|---|---|---|---|---|---|

Prompt: A street with cars lined with poles and wires.

Prompt: A man and two dogs are riding a scooter.

Prompt: Two cats chill in the bathtub, one laying down.

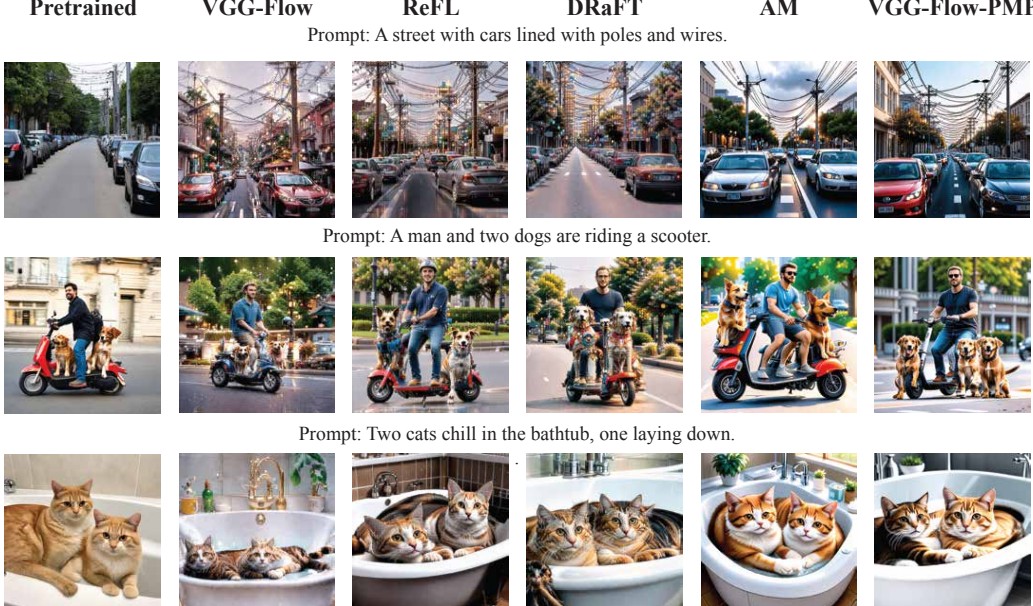

Figure 19: More qualitative results on HPSv2.

| Pretrained | VGG-Flow | ReFL | DRaFT | AM | VGG-Flow-PMP |
|---|---|---|---|---|---|

Prompt: A black lab catching a tennis ball.

Prompt: A tree with blue leaves on a blue hill.

Prompt: Axolotl in the style of minecraft.

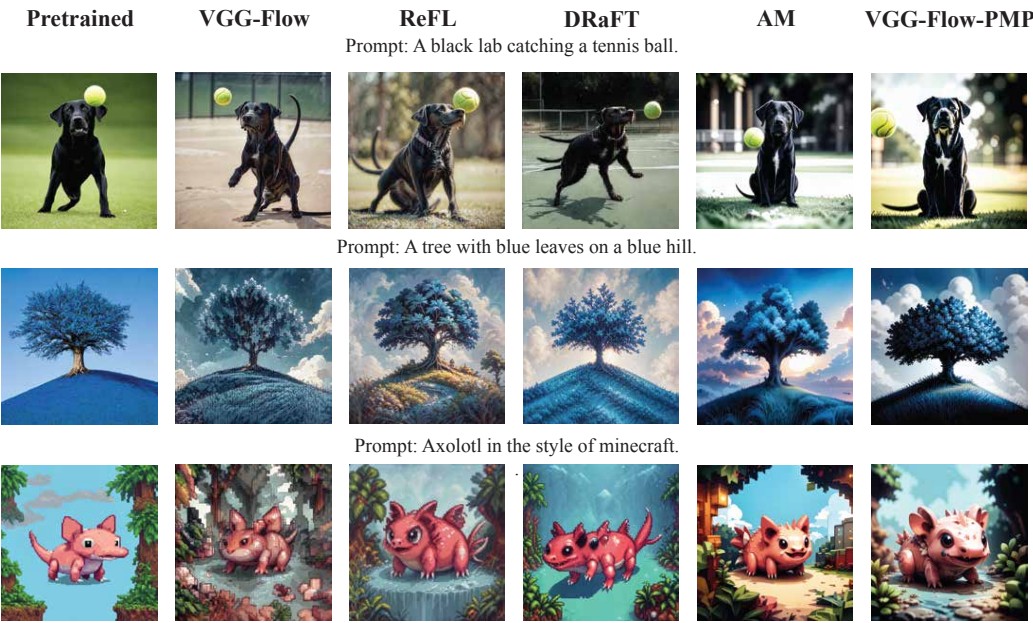

Figure 20: More qualitative results on PickScore.

