# OpenReview forum: "Value Gradient Guidance for Flow Matching Alignment"
_NeurIPS.cc/2025/Conference — NeurIPS 2025 poster_

### Official Review · Reviewer_jDnT · 2025-06-30

**Clarity:** 3
**Significance:** 2
**Originality:** 3
**Rating:** 5
**Confidence:** 4

**Summary:**

The paper proposed a new approach, VGG-Flow, for flow matching alignment. The proposed method was deduced from the theory of optimal control, and effectively learn a vector field network and a value network that approximates the gradient of the running cost simultaneously. On text-to-image generation tasks, the proposed approach exhibited better performance in terms of multiple objective rewards.

**Questions:**

Please refer to the weaknesses listed above. In addition:
- For text-to-image alignment, it is unclear why the CLIP score is missing in the reward. In all previous work, like Adjoint Matching and OCFlow, the CLIP score was regarded as the primary metric due to its robustness to adversarial attacks.
- The network $g$ looks exactly the same as the adjoint state in Adjoint Matching, or the costate in OCFlow. Can you further elaborate on their connections?

**Ethical Concerns:**

["NO or VERY MINOR ethics concerns only"]

**Final Justification:**

Please refer to my discussions with the authors.

**Limitations:**

Yes.

**Paper Formatting Concerns:**

No.

**Quality:**

3

**Strengths And Weaknesses:**

## Strength
- The paper is inspired by the theory of optimal control, which has a solid mathematical background.
- The final algorithm has a clear correspondence to the running cost in the optimal control objective, which offers a fine-tuning method with gradient initialization from the reward function.

## Weakness
- ** Proposition 1, to the best of my knowledge, is incorrect for ODE-based flow matching**. The authors referred to an external source for this proposition. After checking the original paper, I only found the proof for the SDE setup (i.e., diffusion/score matching), whose bound contains the variance of the diffusion (noise) term. Therefore, the adaptation to ODE-based flow matching is non-trivial.
- **The motivations behind the proposed approach seem to focus on the wrong aspect**. For example, the authors claimed that solving the optimal control objective is "intractable". This is clearly an overstatement, as existing approaches like SOC [1] and Adjoint Matching [2] for stochastic flow (SDE), and OCFlow [3] for deterministic flow (ODE) have all proposed practically feasible algorithms for alignment. Instead, the major difference of the proposed approach is that it does not require backpropagating the reward gradient through the ODE/SDE solver, as it learns a model to approximate it. Efficiency (and therefore the practical time benchmark) shall be one of the major contributions.
- Although the authors have noted the close relation to the OC-based method in Section 6, they were never compared as baselines.
- The proposed approach involved more time-consuming operations like JVP and additional training parameters (of the gradient value network). However, a benchmark on the running cost was not provided to give a direct comparison with the existing baselines.
- The results in Table 1 were **unsatisfactory** in demonstrating the superior performance, with **clear errors in highlighting the best model**. For example, HPSv2 reward and FID metrics are erroneously highlighted. The PickScore FID is also incorrect.

[1] Domingo i Enrich, Carles, et al. "Stochastic optimal control matching." Advances in Neural Information Processing Systems 37 (2024): 112459-112504.

[2] Domingo-Enrich, Carles, et al. "Adjoint matching: Fine-tuning flow and diffusion generative models with memoryless stochastic optimal control." arXiv preprint arXiv:2409.08861 (2024).

[3] Wang, Luran, et al. "Training Free Guided Flow Matching with Optimal Control." arXiv preprint arXiv:2410.18070 (2024).

---

> ### Author Rebuttal · Authors · 2025-07-31
>
> We appreciate the reviewer for their time and efforts to review our paper and to raise constructive questions.
>
> > ** Proposition 1, to the best of my knowledge, is incorrect for ODE-based flow matching**. The authors referred to an external source for this proposition. After checking the original paper, I only found the proof for the SDE setup (i.e., diffusion/score matching), whose bound contains the variance of the diffusion (noise) term. Therefore, the adaptation to ODE-based flow matching is non-trivial.
>
> Thank you for pointing this important issue out and we apologize for not carefully checking the cited paper. You are right, this claim is incorrect: without any additional assumptions, we can only claim that the Wasserstein-2 distance is bounded, not the KL divergence. While we lose track of the exact probabilistic formulation of the objective function, finetuning to minimize W2 distance is still a valid setting.
>
> In practice, we may assume that the velocity field $v(x,t)$ is Lipchitz continuous for $t \in [0, 1]$ or $t \in [\epsilon, 1 - \epsilon]$ ($\epsilon > 0$). With this additional mild assumption, we may show that the KL divergence in Proposition 1 is bounded.
>
> We will revise our statements and elaborate the technical details in the next version of our draft.
>
> > The motivations behind the proposed approach seem to focus on the wrong aspect. For example, the authors claimed that solving the optimal control objective is "intractable". This is clearly an overstatement, as existing approaches like SOC [1] and Adjoint Matching [2] for stochastic flow (SDE), and OCFlow [3] for deterministic flow (ODE) have all proposed practically feasible algorithms for alignment. Instead, the major difference of the proposed approach is that it does not require backpropagating the reward gradient through the ODE/SDE solver, as it learns a model to approximate it. Efficiency (and therefore the practical time benchmark) shall be one of the major contributions.
>
> Sorry to possible misunderstanding here: in our paper, we do not claim that OC objectives are intractable (indeed, we have presented empirical results on Adjoint Matching.), but instead argue that our amortized objective is computationally more tractable. We will revise our claims and clearly point out that the major motivation of using HJB instead of ODE/SDE solver is efficiency.
>
> Nevertheless, we would like to point out that we observe that to use Adjoint Matching we need to carefully treat the outliers (for instance, throwing out samples when adjoints get too large) and empirically it does not work that well compared to our method.
>
> Plus, we appreciate the reviewer for mentioning OCFlow and we will cite and explain the connections to OCFlow in our revised draft.
>
> > Although the authors have noted the close relation to the OC-based method in Section 6, they were never compared as baselines.
>
> > The results in Table 1 were unsatisfactory in demonstrating the superior performance, with clear errors in highlighting the best model. For example, HPSv2 reward and FID metrics are erroneously highlighted. The PickScore FID is also incorrect.
>
> > For text-to-image alignment, it is unclear why the CLIP score is missing in the reward. In all previous work, like Adjoint Matching and OCFlow, the CLIP score was regarded as the primary metric due to its robustness to adversarial attacks.
>
> We are sorry that we did not have the chance to present the empirical results on OC-based methods. Our original plan was to show Adjoint Matching (since it is the more relevant and recent one) but it was not open-sourced at that point plus the experiments in the AM paper are done in some different setting.
>
> We apologize for the mistakes in the tables in the main text and thank you for suggesting CLIPScore. Here we present the most recent results after some refinement on the hyperparameters:
>
> **Aesthetic Score**
>
> | Method     | Reward | CLIPScore | DreamSim | FID |
> |------------|--------------|------------|--------------|------------|
> | Ours       | 8.02     | 10.92       | 20.95 | 341 |
> | ReFL       | 10.00     | 5.30       | 5.59  | 1338 |
> | DRaFT       | 9.54     | 7.47       | 7.77 | 1246 |
> | Adjoint Matching       | 6.87    | 15.82       | 22.34 | 465 |
>
> **HPSv2**
>
> | Method     | Reward | CLIPScore | DreamSim | FID |
> |------------|--------------|------------|--------------|------------|
> | Ours       | 3.86     | 15.39       | 18.40 | 1161 |
> | ReFL       |   3.87   | 12.13       | 14.08  | 1195 |
> | DRaFT       | 3.76     | 12.79       | 15.05 | 1177 |
> | Adjoint Matching       | 3.59    | 11.76       | 14.11 | 1247 |
>
> **PickScore**
>
> | Method     | Reward | CLIPScore | DreamSim | FID |
> |------------|--------------|------------|--------------|------------|
> | Ours       | 23.01     | 18.87       | 23.36 | 967 |
> | ReFL       |   23.19   | 15.06       | 17.71  | 997 |
> | DRaFT       | 23.00     | 15.49       | 19.03 | 968 |
> | Adjoint Matching       | 22.78    | 16.12       | 19.69 | 1033 |
>
> > The proposed approach involved more time-consuming operations like JVP and additional training parameters (of the gradient value network). However, a benchmark on the running cost was not provided to give a direct comparison with the existing baselines.
>
> Thank you for raising this question. Please find the runtime table below (running for 400 iterations; 2 GPUs, batch size 64 with gradient accumulation). We would like to note that Adjoint Matching is implemented with a replay buffer while ours is pure on-policy. We may adopt the same trick to accelerate training but we leave that to future work.
>
> Runtime
> | Method     | Time (hours) |
> |------------|--------------|
> | Ours (subsampling 50%)      | 34       |
> | Ours (subsampling 25%)       | 22       |
> | ReFL       |   10       |
> | DRaFT       | 10       |
> | Adjoint Matching          | 14    |
>
> > The network g looks exactly the same as the adjoint state in Adjoint Matching, or the costate in OCFlow. Can you further elaborate on their connections?
>
> They are indeed very closely connected. For smooth optimal control problems, the adjoint states are the same as $g = \nabla V$ **on the optimal trajectory**. Here we would like to underscore that HJB is that it is a globally true condition: even without knowing the optimal trajectories it must hold everywhere. Therefore, one benefit of this HJB formulation is that it gives room for off-policy training.

---

> > ### Comment · Reviewer_jDnT · 2025-08-02
> >
> > I appreciate the authors' detailed rebuttals and comprehensive new experimental results that make the original work more grounded. I also appreciate authors' honesty in acknowledging their (unintentional) mathematical mistakes and their willingness to modify them in the revised manuscript.
> >
> > Overall, I am very satisfied with the rebuttal and the new experimental results, which provide a more comprehensive evaluation of the proposed approach. Even though the results are not always SOTA, I acknowledge the novelty in the proposed approach, as learning a gradient value network, to the best of my knowledge, is indeed a new method that distinguishes this work from the existing families of gradient-based guidance frameworks. Therefore, I decided to raise my overall score from 3 to 5 and also modify the other scores accordingly.
> >
> > The remaining issue is also quite straightforward -- the time complexity is high compared to existing approaches. I agree with the authors' claim that it could be improved in the future. I would further appreciate it if the authors could further elaborate on the detailed methods that could be potentially adapted to improve the running cost.

---

> ### Author Response · Authors · 2025-08-04
>
> We are very glad that we have mostly addressed the reviewer's concerns and we truly appreciate the reviewer's acknowledgement on our contribution.
>
> On methods to save the computational cost:
> - To use replay buffers to save inference time for sampling rollouts, like in every batch only 20% percent are collected from the current policy while the rest are reused from the past.
> - Another very straightforward and feasible method is to have the value gradient network to share layers with the "policy network". It makes sense once we notice that the optimal policy should be the same as value gradients at optimality, plus that the features from the finetune/pretrained model are already good.
> - Empirically, we may explore if detaching some parts of the computational graph may work, just like in temporal difference methods (in RL) people typically use target network which is updated not with backprop but with exponential moving average.

---

### Official Review · Reviewer_Voux · 2025-07-02

**Clarity:** 4
**Significance:** 3
**Originality:** 4
**Rating:** 5
**Confidence:** 4

**Summary:**

This work proposes VGG-Flow, a method for efficiently aligning pretrained flow matching models with reward functions by leveraging optimal control theory. The authors formulate a relaxed control objective whose optimality conditions are given by the Hamilton–Jacobi–Bellman (HJB) equation, showing that the difference between the fine-tuned and base velocity fields should match the gradient of a value function. In practice, they parametrize this value-gradient via a forward-looking approximation, i.e., using the reward gradient of a one-step Euler prediction plus a small learnable residual, and jointly train the velocity field and value-gradient models over simulated trajectories.. Empirically, VGG-Flow is applied to Stable Diffusion 3 under various reward models (e.g., aesthetic score, HPSv2, PickScore) and demonstrates faster increase of rewards while better preserving sample diversity and base-model priors compared to baselines such as ReFL, DRaFT, and Guidance Matching.

**Questions:**

1. I am curious about the choice of the value function network, i.e., using a module initialized from SD v1.5 UNet. Is there any reason why the authors choose this architecture?
2. I think the paper’s most relevant baseline should be adjoint matching, which uses memoryless SDE to derive the value term, and update the velocity. I saw the comparison in supplementary files, but the results are not in the main paper. Is there any reason why adjoint matching is not on the main paper?
3. Furthermore, regarding the results in supplementary, adjoint matching shows better diversity and FID, while other methods such as ReFL or GM shows the highest reward. I can understand that ReFL, GM or DRaFT achieve the highest reward as they are prone to reward hacking. But, when compared to adjoint matching, the better diversity or FID is partly due to the SDE formulation. What do you think about it? If this is true, can the proposed method also be applied with SDE?
4. If one considers multi-reward optimization, how would this be the case? Do we need to compute all the Jacobians for each reward? Also, if we are using VLMs for reward modeling, can the proposed method scale to such a case?

**Ethical Concerns:**

["NO or VERY MINOR ethics concerns only"]

**Final Justification:**

The authors' clarified most of my concerns, I raise my score to 5 accept.

**Limitations:**

They have mentioned the limitations of the work, e.g., the limitations on using first-order gradient estimators, lack of exploration on architectural designs.

**Quality:**

3

**Strengths And Weaknesses:**

### Strength
 The strength of this paper is its theoretically grounded and practical algorithm that introduces the HJB framework, where the method provides a clear probabilistic interpretation and memory-efficient gradient-matching objective. The forward-looking parametrization of the value gradient not only sidesteps costly adjoint ODE solvers but also accelerates adaptation, yielding substantial reward improvements within a tight computational budget. Moreover, the extensive ablations on reward temperature, boundary-loss weighting, and subsampling strategies show the method’s robustness across different reward signals, while quantitative results (e.g., Pareto fronts of reward vs. diversity and FID) convincingly illustrate that VGG-Flow achieves better trade-offs than direct reward-maximization and gradient-matching baselines .

### Weakness
 Despite its elegance, the paper’s theoretical contributions lack rigorous convergence guarantees or bounds on the approximation error introduced by finite-difference estimation of the value gradient, thus relying on heuristic weighting and early-stop selection to avoid reward hacking. Furthermore, technically the method requires approximation of (jacobian of) value function and update the residual with learned value function, which also could induce bias throughout the learning. The empirical evaluation is narrowly focused on a single T2I model and human-preference rewards, while neither multi-objective scenarios (e.g., balancing aesthetics and alignment) nor perceptual user studies are explored, leaving open whether the numeric gains translate to genuinely better image quality.

---

> ### Author Rebuttal · Authors · 2025-07-31
>
> We thank the reviewer for their acknowledgement on our contributions and their time and efforts invested in the reviewing process.
>
> > Despite its elegance, the paper’s theoretical contributions lack rigorous convergence guarantees or bounds on the approximation error introduced by finite-difference estimation of the value gradient, thus relying on heuristic weighting and early-stop selection to avoid reward hacking.
> > Furthermore, technically the method requires approximation of (jacobian of) value function and update the residual with learned value function, which also could induce bias throughout the learning.
>
> Thank you for pointing out the approximation issue. Yes, this is indeed a coarse approximation, though our empirical results show that the finetuning process does not break due to this approximation. In the meantime, we would like to argue that if we aim for fast finetuning, then we have to accept some level of bias unless the cost of carefully finetuning the value function with small learning rates is affordable.
>
> On theoretical guarantees: thank you for your suggestion. We believe that the convergence bound can be shown with classical results on HJB or some recent papers (for instance, [1] on approximation bounds for distributional HJB). Indeed, from the control perspective out setting is simply that we use a different set of costs and dynamics. We will include more analysis and details on this aspect in the next version of our paper.
>
> > The empirical evaluation is narrowly focused on a single T2I model and human-preference rewards, while neither multi-objective scenarios (e.g., balancing aesthetics and alignment) nor perceptual user studies are explored, leaving open whether the numeric gains translate to genuinely better image quality.
>
> We acknowledge that our paper does not explore multi-reward scenarios nor present user studies. We made this decision because 1) our paper focuses on the finetuning method itself rather than the properties of specific reward functions and 2) we follow the same evaluation setting in many other related papers in this domain, such as the baselines (ReFL, DRaFT and Adjoint Matching).
>
> > I am curious about the choice of the value function network, i.e., using a module initialized from SD v1.5 UNet. Is there any reason why the authors choose this architecture?
>
> The choice of this value network is due to the convenience in implementation: it takes few lines to copy the model. Other architectures work for our purpose as well.
>
> > Furthermore, regarding the results in supplementary, adjoint matching shows better diversity and FID, while other methods such as ReFL or GM shows the highest reward. I can understand that ReFL, GM or DRaFT achieve the highest reward as they are prone to reward hacking. But, when compared to adjoint matching, the better diversity or FID is partly due to the SDE formulation. What do you think about it? If this is true, can the proposed method also be applied with SDE?
>
> We argue that this is not due to the SDE formulation. Our method directly applies to the ODE since we always assume $dx = v(x,t)dt$. Adjoint matching, in contrast, turns the ODE into an SDE with the same probability flow because it relies on the assumption of “memoryless” process.
>
> Partially to show why the SDE formulation is not a must, we also show the results of “adjoint matching method without SDE” (see Section A.1 in the supplementary, derived with PMP) in the table below. Even without the equivalent SDE, adjoint matching (or more precisely, optimal control with direct simulation of the adjoint equation) leads to some performance.
>
> (Aesthetic Score)
>
> | Method     | Reward | CLIPScore | DreamSim | FID |
> |------------|--------------|------------|--------------|------------|
> | Adjoint Matching       | 6.87    | 15.82       | 22.34 | 465 |
> | "Deterministic Adjoint Matching"       | 7.53    | 8.26       | 11.17 | 1171 |
>
>
> > If one considers multi-reward optimization, how would this be the case? Do we need to compute all the Jacobians for each reward? Also, if we are using VLMs for reward modeling, can the proposed method scale to such a case?
>
> If we use a linear combination of multi-rewards (e.g., weighted sum), then we do need to compute all the Jacobians as the differentiation of sum equals to sum of differentiations.
>
> Regarding VLM models, it depends on how we utilize the VLM to get a score value. For example, one approach [2] is to ask the the model to give a binary answer about “whether the generated image is aligned with the given text prompt; only answer ‘yes’ or ‘no’”. We may extract the likelihood of the "yes" token. Still, it remains an issue to compute the reward gradient with respect to the input $x_t$. It can possibly be solved by stochastic approximations (e.g., with randomly sampled directions, similar to what Hutchinson's trick is used for estimating divergence).
>
> ---
>
> [1] Tractable Representations for Convergent Approximation of Distributional HJB Equations. Alhosh et al. https://arxiv.org/abs/2503.05563
>
> [2] Beyond Yes and No: Improving Zero-Shot LLM Rankers via Scoring Fine-Grained Relevance Labels. Zhuang et al. NAACL 2024. https://arxiv.org/abs/2310.14122

---

> > ### Comment · Reviewer_Voux · 2025-08-04
> >
> > Thanks for the authors for the response. Regarding comparison with adjoint matching, it seems like SDE formulation is mandatory for adjoint matching, but the proposed algorithm also works for ODE, and it shows comparable performance to adjoint matching. If I am understanding correctly, what are the critical difference between adjoint matching and HJB, that leads to such different behavior?

---

> ### Author Response · Authors · 2025-08-04
>
> We thank the reviewer for their response. Below we discuss the key properties and assumptions made in each formulation: the original adjoint matching, the "deterministic" variant (derived with Pontryagin's maximum principle in our paper), and our HJB formulation.
>
> ## Why Adjoint Matching uses SDE?
>
> We would also like to first elaborate a bit more on why the AM paper turns ODEs into SDEs. In Equation 23 of the AM paper, the authors write that, with a SDE equivalent to the ODE (in the probabilistic flow sense), what the optimal $p(X_0, X_1)$ ($X_0, X_1$ are the starting and end points of the SDE) is with the finetuning objective $p_\text{base}(X) \exp(r(X)) / Z$. Specifically, it is
>
> $$p^*(X_0, X_1) = p_\text{base}(X_0, X_1)\exp(r(X_1) + V(X_0, 0))$$
>
> where $V(X_0, 0)$ is the value function of $X_0$ at time $t=0$. The optimal distribution of $X_1$ is therefore unfortunately coupled with $X_0$ in general, deviating from the ideal finetuning objective.
>
> The authors discover that only with the memoryless schedule that one can decompose $p_\text{base}(X_0, X_1)$ into $p_\text{base}(X_0)p_\text{base}(X_1)$, which subsequently leads to the ideal finetuning objective (see Equaion 24 in the AM paper).
>
> Compared to the AM setting, **we formulate the finetuning objective as a relaxed one**, since without any additional assumptions, we essentially replace the KL divergence term in the expectation bracket in Equation 1 of our paper
>
> $$-r(x_1) + \lambda \text{KL}(p_\theta(x_1) \|\| p_\text{base}(x_1))$$
>
> with the Wasserstein-2 distance. With additional assumptions, our formulation can be seen as optimizing some upper bound of the KL divergence (see our response to Review jDnT). Due to this difference, we may use our HJB-based method and also the "deterministic" adjoint matching directly on ODEs. For more details, please see our discussions on Pontryagin's maximum principle and the appendix in our paper.
>
> ## Key techinical differences between adjoint matching and the "deterministic" variant
>
> Beyond the difference in objectives, we perceive some other critical differences between the original AM and the "deterministic" one:
>
> - the additional variance of model update gradients due to the noise of SDE sampling and,
> - the numerical issues from pretrained flow matching models to the equivalent SDE.
>
> For the first point, we believe it is not that surprising and probably the major reason for the difference in empirical results. For the second one, we argue that, while the SDE is theoretically equivalent to the original ODE in expectation, a flow matching model trained on real-world high-dimensional data is hardly perfect. Training with points that are sampled with unexpected noise may lead to degradation in performance.
>
>
> ## HJB vs. AM
>
> We first note the important fact that HJB equation must hold for any point in the space while the definition of adjoints only make sense on the optimal trajectories.
>
> Our HJB formulation has the following advantages: it
>
> - can leverage the one-Euler-step guess $\hat{x}_1 = x_t + (1-t)v(x,t)$ to parameterize $\nabla V = -\nabla r(\hat{x}_1) + \text{nn}(x_t, t)$ for faster convergence
> - allows for amortized and off-policy training (since HJB must hold everywhere), and
> - introduces the value gradient network that can potentially absorb some of the outliers bad for optimization (while in the official implementation of adjoint matching, outliers with adjoint magnitude too large are thrown away)
>
> ---
> We hope the above may address some of the reviewer's concerns.

---

> > ### Comment · Reviewer_Voux · 2025-08-08
> >
> > Thanks for the detailed response! I also believe that such details on the difference with respect to adjoint matching should be included in the final version.

---

### Official Review · Reviewer_UNtT · 2025-07-02

**Clarity:** 2
**Significance:** 2
**Originality:** 3
**Rating:** 4
**Confidence:** 2

**Summary:**

The authors consider a composite loss to finetune Flow Matching models following a reward function designed to better align generated images with human preferences.
This loss is a trade-off between a $\ell_2$ term that keeps the fine-tuned model closed to the pretrained velocity field (the prior model)  and a term that maximizes in expectation the reward on the generated images.
To circumvent the need to generate images in the loss (which involves solving the ODE), the authors leverage optimal control theory: they instead use a surrogate network that models the gradient of the value function in the Hamilton-Jacobi-Bellman (HJB) equation.

**Questions:**

First, I think the paper deserves a better positioning in the literature:
- One motivation stated by the paper is that there is a strong difference between diffusion and flow matching in order to leverage OC. The sentence "the key challenge" l25 should be clarified.
- I think the introduction does not clearly present the goal of the paper. As I understand, it is to avoid methods both offline RL methods  (where one would need the pretrained training set) or online (where one would need to generate new samples on the fly).

**On the preliminaries on FM models** There seems to be an error in $u(x_t | x_1)$. I guess it should be $\frac{x_1 - x_t}{1-t}$. I also wonder why the authors specify a variant (rectified flows) so early while some more crucial details are missing for a clear introduction to FM: define how $x_t$ is built (linear interpolation $x_t = (1-t)x_0 + tx_1$ ?).
I understand the need for brevity but I think this subsection could be handled a bit more carefully, especially given some confusion on $u(x_t |x_1)$ seem to arise again later in the paper (l204: the factors are now inverted).

**On the preliminaries on Optimal Control**
- As far as I understand, the use of optimal control theory in this ODE FM setting is the main highlight of the paper: then, this subsection is important and has to be more pedagogical.  Notably, when writing the continuity equation as a HJB equation: precisely define what is $V$ ($- \log p_t$ ? ), the control signal, ... or provide a reference.
- The sentence from l.121 to l.123 is  unclear: is it that the CE does not enable to access the $p_t$ given a pretrained velocity field or is it taht the maximum likelihood is not a good objective to train a velocity field  (due to the infinite number of paths, from which comes Flow Matching which explicitly regress against a given conditional velocity field).

**4.1: The method**
There are still many points that remain unclear.
- I think the setup could be more rigorously exposed (differentiable reward function, etc).
- While the original loss looks very natural and looks quite common (similar to [20]), I have trouble understanding how the reparameterization through the HJB equation makes things computationally easier (if that is the goal). How can training a new model (which is also a Stable Diffusion UNET) can make things go quicker than generating samples (with OT/rectified FM models, this can be achieved with few Euler steps as noted by the authors themselves). I feel like that while being elegant, it introduces the need for many approximations afterwards, one of them being that at the end you still generate points (but you do it with one Euler step).

**On the experiments**  In Table 1, it seems that concurrent models achieve lower FID which is desirable and yet are not put in bold characters.

- While the visual results look quite convincing, there is always this trading off between the two terms in the loss (the one maximizing the reward and the one forcing to stay close to the base model): I would like to see of this parameter impacts the metrics, and same for concurrent methods that must have some kind of similar tradeoff.
 As it is in the paper, it is still unclear the impact of each parameter on the tradeoff (is it $\beta$, $\lambda$, $\alpha$, $\eta_t$: especially given by Figure 6 where there seems to be a discrepancy between the $\beta$ in the caption and the $\lambda$ in the legend ?)

- The main missing point for me regarding the experiments is whether you achieve faster training time (you gain in terms of iterations but what about computation time ?).  Among the benchmarks models you pick, is there any methods where images are generated (since, as i understand you claim to circumvent this aspect).

**Ethical Concerns:**

["NO or VERY MINOR ethics concerns only"]

**Final Justification:**

I raised my score to weak accept, as the authors have provided a strong rebuttal, with additionnal ablation studies and explanations on their method. I believe the interest of the paper lies in the new use of the HJB equation for FM alignment with a given reward function. I do not give a higher score because the practical implementation of the method still relies on several approximations.  Moreover, it does not eliminate the need for empirical tuning to balance reward maximization and sample quality.

**Limitations:**

Yes

**Quality:**

2

**Strengths And Weaknesses:**

**Strengths**

The overall structure of the paper is quite clear and easy-to-follow. The writing style is good.
The fact that the paper takes its roots in the optimal control theory is quite elegant.

**Weaknesses**

The point that puzzles me the most is the additional complexity the method seems to require: it involves an extra network jointly trained in the loss. This then imply many approximations and extra hyperparemeters to maintain the method tractable (e.g. parameterization of $g_\phi$, stop gradient, finite order methods), while the initial objective doesn't seem at first overly complex. See questions below for more details.

---

> ### Author Rebuttal · Authors · 2025-07-31
>
> We appreciate the reviewer for their time and efforts in reviewing our paper and their constructive questions.
>
> > ...many approximations and extra hyperparameters to maintain the method tractable (e.g. parameterization of g_phi, stop gradient, finite order methods), while the initial objective doesn't seem at first overly complex.
>
> Thank you for raising this issue. We do agree that there are probably engineering details in implementing our method. In the meantime, we would like to highlight that the initial objective is not an easy one: tracking the exact value function (which indicates the probability flow) itself is hard. In discrete time, obtaining the value function is often done with Bellman equation / temporal difference (TD) methods and people have designed many tricks to stabilize RL training: target network (with gradients detached) with exponential moving average, trust region / KL clip (as in PPO), replay buffer, etc. The continuous optimal control shares the same nature as the discrete counterpart and therefore cannot be much simpler. In addition, the HJB equation involves derivatives in high dimensional spaces and therefore some approximation has to be done there (another potential approach is that we turn this problem back to the discrete-time RL problem).
>
> > One motivation stated by the paper is that there is a strong difference between diffusion and flow matching in order to leverage OC. The sentence "the key challenge" l25 should be clarified.
>
> We apologize for this vague statement. We meant that in diffusion models, we have the (approximate) score function $s_\theta(x_t, t) = \nabla \log p_t(x_t)$ and therefore performing distribution matching (with the reward function) is relatively easy. For flow matching models which is trained by matching the velocity $v_\theta(x_t,t)$ with some predefined reference path $u(x_t; x_0)$, the "probability" of a state $(x_t, t)$ is not obvious for general flow matching models (for instance, models other than rectified flow models). It is even harder in the case that we are only given model weights but not how it is trained. This is why we turn to the optimal control formulation and solve for the value function (that indicates the probability flow) in an online manner.
>
> On rectified flow models: for a pretrained rectified flow model, we know an easy way to compute probability flow $p_t(x_t)$; however, once this model is finetuned without a dataset, the model starts to deviate from being a rectified flow model and the probability flow in theory cannot be computed that way.
>
> > I think the introduction does not clearly present the goal of the paper. As I understand, it is to avoid methods both offline RL methods (where one would need the pretrained training set) or online (where one would need to generate new samples on the fly).
>
> Sorry for the confusion. The purpose of the paper is to find an online and computationally-efficient RL method that finetunes any flow matching model with any differentiable reward model. Offline methods typically assume the presence of trajectory data for training, while here we only have a reward function without any dataset given.
>
> > On the preliminaries of FM method.
>
> We apologize for the typo and will fix it in later versions.
>
> > On the preliminaries on Optimal Control. As far as I understand, the use of optimal control theory in this ODE FM setting is the main highlight of the paper: then, this subsection is important and has to be more pedagogical. Notably, when writing the continuity equation as a HJB equation: precisely define what is V(-log pt), the control signal, ... or provide a reference.
>
> We will provide more pedagogical details about the foundation of optimal control in Sec 3.2 in the final version. $V$ is the value function of the control problem, which we define in Eq. 5. Currently, we refer to the Sec 4.1 of [1] paper for a self-contained preliminary of optimal control formulation.
>
> > The sentence from l.121 to l.123 is unclear: is it that the CE does not enable to access the pt given a pretrained velocity field or is it that the maximum likelihood is not a good objective to train a velocity field (due to the infinite number of paths, from which comes Flow Matching which explicitly regress against a given conditional velocity field).
>
> Thank you for raising this point and sorry for the unclear explanation here. We wanted to mention that with only terminal rewards or distributions, it is typically hard to train a model and the L2 running cost (the soft constraint between the finetuned and pretrained models) can be seen as a component to stabilize training. This is similar to diffusion models and flow matching models where a noising process or a reference velocity field is used. We will polish the sentence and explain the intuition in a clearer way.
>
> > 4.1: There are still many points that remain unclear. I think the setup could be more rigorously exposed (differentiable reward function, etc).
>
> Thanks for pointing this out and we will explicitly explain the setup in the last two paragraphs in the intro section and the beginning of the method section.
>
> On our setting:
>
> Given a differentiable reward function, we want to train our generative model to achieve high reward scores for the generated samples and also to preserve the prior distribution of the pretrained model. To be more specific, we formulate this to Eq. 9 via the optimal control framework and then focus on solving the control problem. We refer to our Sec 4.1 for a description of the setting, which we will make sure to refine in our final version.
>
> > ...how the reparameterization through the HJB equation makes things computationally easier (if that is the goal)..
>
> Good question! Here are our intuitions:
>
> - By using the HJB formulation, we know that the optimal velocity is simply the value gradient (up to a sign). The value function itself can be well approximated by single-Euler-step guess. The additional U-Net is used to correct the errors in this approximation so that the model, after long periods of training, is not too biased.
> - Optimizing the value function in an amortized way (with some approximations like finite differences) gives us benefits in computational efficiency: instead of solving for the adjoints, we may subsample transitions for training, especially in the case when the number of inference steps is large. This is similar to the motivation of using temporal difference (TD) methods in RL (for instance, soft actor-critic [1]) instead of tracking a very long trajectory.
>
> > On the experiments In Table 1, it seems that concurrent models achieve lower FID which is desirable and yet are not put in bold characters.
>
> Sorry for this mistake. We indeed have revised our experiments by setting the subsampling rate in our method to 50% (previously 25%). Please fine the table below:
>
> **Aesthetic Score**
>
> | Method     | Reward | CLIPScore | DreamSim | FID |
> |------------|--------------|------------|--------------|------------|
> | Ours       | 8.02     | 10.92       | 20.95 | 341 |
> | ReFL       | 10.00     | 5.30       | 5.59  | 1338 |
> | DRaFT       | 9.54     | 7.47       | 7.77 | 1246 |
> | Adjoint Matching       | 6.87    | 15.82       | 22.34 | 465 |
>
> **HPSv2**
>
> | Method     | Reward | CLIPScore | DreamSim | FID |
> |------------|--------------|------------|--------------|------------|
> | Ours       | 3.86     | 15.39       | 18.40 | 1161 |
> | ReFL       |   3.87   | 12.13       | 14.08  | 1195 |
> | DRaFT       | 3.76     | 12.79       | 15.05 | 1177 |
> | Adjoint Matching       | 3.59    | 11.76       | 14.11 | 1247 |
>
> **PickScore**
>
> | Method     | Reward | CLIPScore | DreamSim | FID |
> |------------|--------------|------------|--------------|------------|
> | Ours       | 23.01     | 18.87       | 23.36 | 967 |
> | ReFL       |   23.19   | 15.06       | 17.71  | 997 |
> | DRaFT       | 23.00     | 15.49       | 19.03 | 968 |
> | Adjoint Matching       | 22.78    | 16.12       | 19.69 | 1033 |
>
> > While the visual results look quite convincing, there is always this trading off between the two terms in the loss (the one maximizing the reward and the one forcing to stay close to the base model): I would like to see of this parameter impacts the metrics, and same for concurrent methods that must have some kind of similar tradeoff.
>
> We apologize that we accidentally use $\beta = 1 / \lambda$ to indicate the trade-off between reward maximization and prior preservation, for which the results are shown in Figure 6. For $\eta_t$, please find the results on our new ablation study on Aesthetic Score below:
>
> | Method     | Reward | CLIPScore | DreamSim | FID |
> |------------|--------------|------------|--------------|------------|
> | $\eta_t = (1-t)^2$   |  8.02     | 10.92       | 20.95 | 341 |
> | $\eta_t = (1-t)$       | 8.13     | 10.06       | 17.75  | 484 |
>
> > ...whether you achieve faster training time (you gain in terms of iterations but what about computation time ?). Among the benchmarks models you pick, is there any methods where images are generated (since, as i understand you claim to circumvent this aspect).
>
> Please find the runtime table below (running for 400 iterations; 2 GPUs, batch size 64 with gradient accumulation). We would like to note that Adjoint Matching is implemented with a replay buffer while ours is pure on-policy. We may adopt the same trick to accelerate training but we leave that to future work.
>
> | Method     | Time (hours) |
> |------------|--------------|
> | Ours (subsampling 50%)      | 34       |
> | Ours (subsampling 25%)       | 22       |
> | ReFL       |   10       |
> | DRaFT       | 10       |
> | Adjoint Matching          | 14    |
>
> ---
> [1] Adjoint Matching: Fine-tuning Flow and Diffusion Generative Models with Memoryless Stochastic Optimal Control. Domingo-Enrich et al. ICLR 2025.
>
> [2] Soft Actor-Critic: Off-Policy Maximum Entropy Deep Reinforcement Learning with a Stochastic Actor. Haarnoja et al. ICML 2018.

---

> > ### Comment · Reviewer_UNtT · 2025-08-04
> > **Response to authors**
> >
> > I would like to thank the authors for their extensive and thoughtful response, as well as for the new ablation studies.
> > I will raise my score accordingly. That said, I believe a stronger pedagogical effort is really needed in the introduction, particularly regarding the intro to FM and OC. These are not new concepts, and they could be presented more clearly.
> >
> > I believe the main interest of the paper lies in its elegant new theoretical perspective based on optimal control. On the practical side, however, I remain somewhat skeptical, as training an additional auxiliary model appears costly and introduces new hyperparameters and several layers of approximation, even if I understand that such practices are common in reinforcement learning.

---

> > > ### Author Response · Authors · 2025-08-04
> > >
> > > We are very glad that most of the reviewer's concerns are addressed, and we greatly appreciate the reviewer for their acknowledge on our contribution. We also thank the reviewer for pointing out the needs for more pedagogical explanation of key concepts and we will surely improve our draft according to the suggestions of the reviewer.
> > >
> > > Additional comments to answer the reviewer's lingering concerns:
> > >
> > > - Why our method can converge that fast if we need to learn an additional model? It is because that we leverage the educated guess on the value gradient with the one-Euler-step prediction: $\hat{x}_1 = x_t + (1-t) v(x,t)$ -- a guess that can be accurate enough in state-of-the-arts flow matching models. This guess leads to our parameterization of the value gradient (with respect to $x_t$): $\nabla V = -\nabla r(\hat{x}_1) + h(x_t, t; \phi)$. With this guess, we only have to learn the residual value gradient field $h(\cdot, \cdot; \phi)$ which is relatively easy.
> > > - On the extra computational cost: we listed a few possible methods to improve efficiency in our following-up response to Reviewer jDnT.

---

### Official Review · Reviewer_FPr9 · 2025-07-03

**Clarity:** 4
**Significance:** 3
**Originality:** 2
**Rating:** 5
**Confidence:** 2

**Summary:**

This paper proposes VGG-Flow, a method for aligning flow matching models with reward models using a value gradient derived from optimal control theory. The approach estimates a value gradient via the Hamilton-Jacobi-Bellman equation and uses it to guide residual velocity updates. Experiments show improvements over existing baselines in reward alignment while preserving prior and diversity.

**Questions:**

Can the authors clarify how sensitive the method is to the choice of reward model? For instance, do different reward functions lead to different optimization dynamics or mode collapse rates?


How does the proposed method behave when the reward model is noisy or weakly aligned with human judgment? Could this be mitigated through smoothing or ensemble strategies?

**Ethical Concerns:**

["NO or VERY MINOR ethics concerns only"]

**Final Justification:**

Thank you to the authors for the clear and detailed rebuttal. The explanations regarding the availability of reward models, the theoretical basis of the trade-off parameter, and the principled derivation from optimal control addressed my main concerns. These clarifications improved my understanding and confidence in the work. While some questions remain (e.g., robustness to weak reward models), I believe the core contribution is solid, and I have raised my score accordingly.

**Limitations:**

The method assumes access to pretrained, differentiable, and trustworthy reward models, which may not always be available in other domains (e.g., text, video).

Although the proposed approach is efficient, its performance on much larger models (e.g., >7B parameters) is not yet demonstrated.

**Quality:**

3

**Strengths And Weaknesses:**

Strengths:
- The paper proposes a principled formulation based on deterministic optimal control, which is novel in the context of aligning flow matching models.
- The idea of parametrizing and matching value gradients offers a computationally efficient alternative to adjoint-based methods.
- Empirical results on reward score, sample diversity, and prior preservation are solid and consistently outperform baselines.

Weaknesses:
- The method assumes access to a pretrained reward model, which may be hard to obtain or validate in real-world settings. The paper does not discuss the impact of reward model quality on the results.
- The gradient-matching framework is conceptually similar to existing diffusion alignment methods; the novelty lies mainly in adapting it to flow matching models.
- While the proposed method preserves prior better than other baselines, the trade-off between reward maximization and sample quality remains heuristic, and more theoretical or empirical analysis of this tension would strengthen the work.

---

> ### Author Rebuttal · Authors · 2025-07-31
>
> We appreciate the reviewer for their acknowledgement on our contribution and raising their concerns.
>
> > The method assumes access to a pretrained reward model, which may be hard to obtain or validate in real-world settings. The paper does not discuss the impact of reward model quality on the results.
>
> > The method assumes access to pretrained, differentiable, and trustworthy reward models, which may not always be available in other domains (e.g., text, video).
>
> We argue that the pretrained models, even the differentiable ones, are common and relatively easy to obtain these days. Pretrained reward models (with differentiable neural nets) are prevalent in the field of foundation models, and many of them are open-sourced and available on HuggingFace. There are publicly available preference datasets (such as HPDv2) with more than 100k text-image pairs. People from the industry are working on larger scale datasets; for instance, the CVPR best paper “Rich human feedback for text-to-image generation” [1] builds a pixel-aligned dense reward model (i.e., they have a map of rewards, not just one single scalar) from a large-scale dataset. On training these reward models: people typically assume the Bradley–Terry preference model, which turns out to be a simple logistic regression problem: telling which one is preferred in a pair of data points with binary classification. Beyond these models, we may also adopt pretrained VLMs to extract differentiable likelihood values (e.g., [2]) as rewards.
>
> Our paper aims to build a generic RL finetuning method that works on any given differentiable reward model and therefore we believe a detailed discussion on reward models themselves are beyond the scope of this particular work.
>
> > The gradient-matching framework is conceptually similar to existing diffusion alignment methods; the novelty lies mainly in adapting it to flow matching models.
>
> The resulting algorithm indeed seems similar to other existing works in high level, and one can surely design such heuristics and try them out. The key differentiating factors of our paper are that 1) our method is derived with a principled approach from an optimal control perspective where the gradient of value function is naturally the velocity, 2) we show what the correction term for the naive heuristics should be.
>
> > While the proposed method preserves prior better than other baselines, the trade-off between reward maximization and sample quality remains heuristic, and more theoretical or empirical analysis of this tension would strengthen the work.
>
> Thank you for pointing it out. We model the finetuning objective in Equation 9 in line 134 in which the $\lambda$ parameter controls the trade-off between reward following and prior preservation. We accidentally used $\beta=1/\lambda$ to indicate this trade-off in the ablation study (Figure 6).
>
> From a mathematical side, the $\beta=1/\lambda$ parameter on the reward model (from Eq. 9), to some extent, is similar to the classifier-free guidance scale (typically set to 7.5 or 5.0) which is something heuristic even at this point. Similar to prior works such as adjoint matching, when $\beta$ is large, the method focuses more on reward maximization and vice versa. The trade-off is inevitably empirical. One may prefer better reward following instead of sample quality/prior. For instance, if one aims for generating sci-fi looking images, then imposing a very strong prior of natural images makes less sense.
>
>
> > Can the authors clarify how sensitive the method is to the choice of reward model? For instance, do different reward functions lead to different optimization dynamics or mode collapse rates?
>
> Empirically we do not find the model very sensitive to reward models, though the margin compared to other baseline methods may vary. For instance, we observed that finetuning with Aesthetic Score is prone to overfitting and our method is more robust and more efficient; on HPSv2 and PickScore, the baseline methods do not suffer that much.
>
> > How does the proposed method behave when the reward model is noisy or weakly aligned with human judgment? Could this be mitigated through or ensemble strategies?
>
> A prior work [3] does apply reward smoothing (by averaging the rewards of sampled neighboring points) due to concerns about the non-smoothness of the reward function.
> In our work we explored this trick in an early stage but did not find it useful in our setup. We have not explored ensemble strategies because we focus on the setting where a fixed reward function is given. That being said, DRaFT (one of the baselines, [4]) has an interesting experiment where they try to use a linear combination of different reward functions to achieve a mixed effect.
>
>  ---
> [1] Rich Human Feedback for Text-to-Image Generation. Liang et al. CVPR 2024. https://arxiv.org/abs/2312.10240
>
> [2] Beyond Yes and No: Improving Zero-Shot LLM Rankers via Scoring Fine-Grained Relevance Labels. Zhuang et al. NAACL 2024. https://arxiv.org/abs/2310.14122
>
> [3] Efficient Diversity-Preserving Diffusion Alignment via Gradient-Informed GFlowNets. Liu et al. ICLR 2025. https://arxiv.org/abs/2412.07775
>
> [4] Directly Fine-Tuning Diffusion Models on Differentiable Rewards. Clark et al. ICLR 2024. https://arxiv.org/abs/2309.17400

---

> > ### Comment · Reviewer_FPr9 · 2025-08-09
> >
> > Thank you to the authors for the clear and detailed rebuttal. The explanations regarding the availability of reward models, the theoretical basis of the trade-off parameter, and the principled derivation from optimal control addressed my main concerns. These clarifications improved my understanding and confidence in the work. While some questions remain (e.g., robustness to weak reward models), I believe the core contribution is solid, and I have raised my score accordingly.

---

### Decision · Program_Chairs · 2025-09-17

**Decision:**

Accept (poster)

**Comment:**

Reviewers agree that this is a significant step forward in the field. The rebuttal and post-rebuttal discussion clarified initial questions. Congratulations, this submission is accepted.